# Quantile Activation: Correcting a failure mode of traditional ML models

**Aditya Challa**                                        *adityac@goa.bits-pilani.ac.in*
*Department of CS&IS and APPCAIR*
*BITS Pilani KK Birla Goa Campus, India*

**Sravan Danda**                                         *dandas@goa.bits-pilani.ac.in*
*Department of CS&IS and APPCAIR*
*BITS Pilani KK Birla Goa Campus, India*

**Laurent Najman**                                       *laurent.najman@esiee.fr*
*Department of CS Univ Gustave Eiffel, CNRS, LIGM, F-77454*
*Marne-la-Vallée, France*

**Snehanshu Saha**                                       *snehanshus@goa.bits-pilani.ac.in*
*Department of CS&IS and APPCAIR*
*BITS Pilani KK Birla Goa Campus, India*

## Abstract

Standard ML models fail to infer the context distribution and suitably adapt. For instance, the learning fails when the underlying distribution is actually a mixture of distributions with contradictory labels. Learning also fails if there is a shift between train and test distributions. Standard neural network architectures like MLPs or CNNs are not equipped to handle this.

In this article, we propose a simple activation function, quantile activation (QAct), that addresses this problem without significantly increasing computational costs. The core idea is to "adapt" the outputs of each neuron to its *context distribution*. The proposed quantile activation (QAct) outputs the relative quantile position of neuron activations within their context distribution, diverging from the direct numerical outputs common in traditional networks.

A specific case of the above failure mode is when there is an inherent distribution shift, i.e the test distribution differs slightly from the train distribution. We validate the proposed activation function under covariate shifts, using datasets designed to test robustness against distortions. Our results demonstrate significantly better generalization across distortions compared to conventional classifiers and other adaptive methods, across various architectures. Although this paper presents a proof of concept, we find that this approach unexpectedly outperforms DINOv2 (small), despite DINOv2 being trained with a much larger network and dataset.

## 1   Introduction

Thanks to deep learning approaches, machine learning has been adopted across wide variety of domains. However, there is a significant failure mode within the standard framework of machine learning. Since, the functions within standard hypothesis classes are fixed, they cannot *adapt* to the distributions (both at train and test time).

**Failure mode of ML Systems:**  Standard ML models assume that the samples are i.i.d drawn from a common distribution $p(x, y)$, and accordingly assume a fixed set of hypothesis $\mathcal{H}$. This is not a realistic

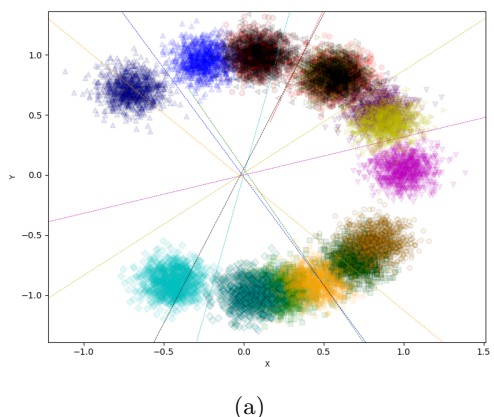

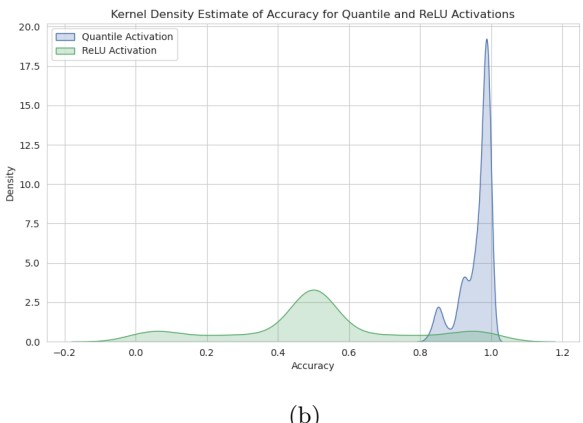

(a)                                                      (b)

Figure 1: A simple toy example to illustrate where ML systems fail. (a) The distribution is a mixture of Gaussian distributions whose centers $(\mu_1, \mu_2)$ are separated by $30°$. The centers themselves can lie anywhere on the unit circle. (please refer to the text for exact description) The dotted lines indicate the optimal linear classifier for a given $\mu_1, \mu_2$, across different values of $\mu_1, \mu_2$.. (b) Histogram of accuracy over 1000 different combinations of $\mu_1, \mu_2$ for both ReLU activation and after incorporating QAct. Clearly, ReLU activation alone cannot perform better than random guess. Incorporating QAct on the other hand can easily infer the latent $\mu_1, \mu_2$.

assumption. In practice, samples are drawn from different (but related) distributions $\{p_i(\mathrm{x}, \mathrm{y})\}$ both at train and test time. And any function class which cannot adapt fails in this scenario. This failure mode manifests itself in a lot of different ways.

**A simple toy example to illustrate the failure mode of ML systems:** To concretely illustrate this failure mode, we present a toy example where samples can potentially have contradictory labels. Consider the distribution generated as follows (figure 1a):

$$
\begin{aligned}
&\mu_1 \sim \mathcal{U}(S^1) && \text{random sample from uniform distribution on the circle} \\
&\mu_2 = R(30°)\mu_1 && \mu_2 \text{ is obtained by rotating } \mu_1 \text{ by } 30° \\
&\text{Class } 0 \sim \mathcal{N}(\mu_1, 0.1\boldsymbol{I}) && \text{Class 0 generated using normal with mean } \mu_1 \text{ and stdev 0.1} \\
&\text{Class } 1 \sim \mathcal{N}(\mu_2, 0.1\boldsymbol{I}) && \text{Class 1 generated using normal with mean } \mu_2 \text{ and stdev 0.1}
\end{aligned} \tag{1}
$$

*Why is this classification problem hard?* Note that any point $x$ from the support of the above distribution is equally likely to be class 0 or class 1. So, one cannot construct any *fixed* function depending only on the input features $x$. Thus, most of existing ML frameworks, which insist on learning a fixed function, fail. Nevertheless, this is a valid distribution where classification with accuracy $\approx 1$ is theoretically possible when one can reconstruct the latent $\mu_1, \mu_2$.

Specifically, the current neural network architectures fail as well. Consider training a simple MLP on this dataset using gradient descent. That is, each batch (of size $B$) of samples is generated as - Sample $\mu_1, \mu_2$, then sample $B/2$ points each from $\mathcal{N}(\mu_1, 0.1\boldsymbol{I})$ and $\mathcal{N}(\mu_2, 0.1\boldsymbol{I})$ respectively. Since, we would have that a specific sample is equally likely to belong class 0 or 1, one would learn (probability) $p = 0.5$ for all the samples. This is verified in figure 1b where we use ReLU activation.

**Fixing the failure mode and defining the "context":** This experiment demonstrates that traditional activation functions like ReLU fundamentally cannot resolve contradictory labels in mixed distributions. The root cause lies in the static nature of existing 'go-to' activations. If instead, neurons could adapt to their *context distribution* - defined by the batch of samples - we could overcome this limitation. This leads us to a novel activation, QAct which uses context distribution to allow individual neurons to adapt. To the best of our knowledge current neural network architectures with ReLU activations such as MLP/CNN do

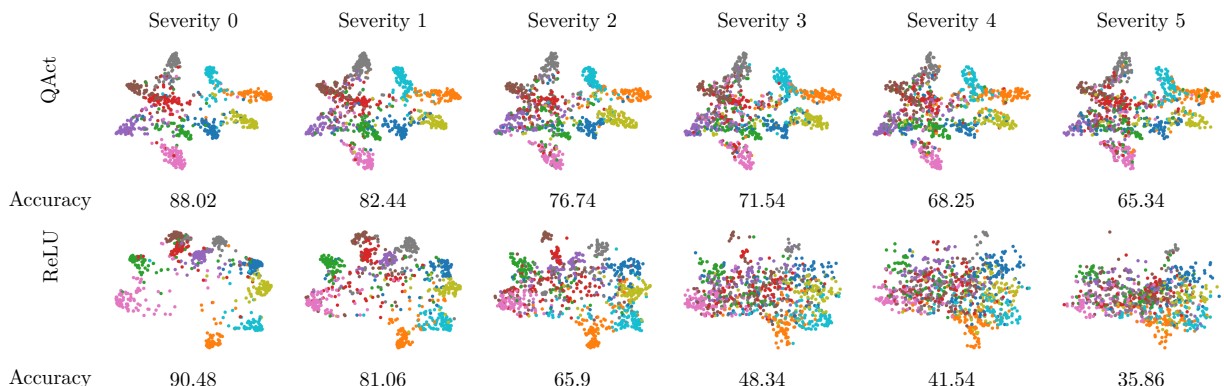

Figure 2: Comparing TSNE plots of QAct and ReLU activation on CIFAR10C with Gaussian distortions. Observe that QAct maintains the class structure extremely well across distortions, while the usual ReLU activations loses the class structure as severity increases.

not consider this. While transformers consider this to some extent via the self-attention module, it is still sample specific and is very expensive computationally to obtain self-attention for the entire batch. Moreover, vision transformers do not consider attention across different samples in the batch.

**Quantile Activation can identify the context distribution from each batch and fix the failure mode** : The key idea is that – at each neuron, the final activation value depends on the activations of the entire batch of samples. Specifically, we use the relative quantile of the pre-activation [1] with respect to other pre-activations in the batch. (Details in section 2). Figure 1b shows that this simple change can allow the neural network to learn in spite of the contradictory labels.

**Distribution Shift as an example of failure mode:** Notably, this failure mode extends beyond synthetic examples. A critical real-world manifestation occurs in distribution shifts where the training is done on a distortion-free distribution while testing happens on distorted distributions. We validate the proposed quantile activation on the standard distribution shift dataset - CIFAR10C. Figure 2 illustrates the results obtained using ReLU activations and QAct. As severity increases (w.r.t Gaussian Noise), we observe that ReLU activation loses the class structure. On the other hand, the proposed QAct framework does not suffer from this and the class structure is preserved with QAct.

## 1.1 Contributions

In (Challa et al., 2024b), the authors propose an approach to calibrate a pre-trained classifier $f_\theta(\boldsymbol{x})$ by extending it to learn a *quantile function*, $Q(\boldsymbol{x}, \theta, \tau)$ ($\tau$ denotes the quantile), and then estimate the probabilities using $\int_\tau I[Q(\boldsymbol{x}, \theta, \tau) \geq 0.5] d\tau$ [2]. They show that this results in probabilities which are robust to distortions.

1. In this article, our approach pivots to the level of a neuron, by suitably deriving the forward and backward propagation equations required for learning (section 2).
2. We then show that a suitable incorporation of our extension produces context dependent outputs at the level of each neuron of the neural network.
3. Steps 1 and 2 in our approach are non-trivially different from (Challa et al., 2024b) and thus contribute to achieving better generalization across distributions and is more robust to distortions, across architectures. We evaluate our method using different architectures and datasets, and compare with the current state-of-the-art [3] – DINOv2(small). We show that QAct proposed here is more robust to distortions than DINOv2, even if we have considerably less number of parameters (22M

---

[1] We use the following convention – "Pre-activations"– which denote the inputs to the activation functions and "Activations" denote the outputs of the activation function

[2] $I[]$ denotes the indicator function

[3] within comparable model sizes

for DINOv2 vs 11M for Resnet18). Additionally, DINOv2 is trained on 20 odd datasets, before being applied on CIFAR10C; in contrast, our framework is trained on CIFAR10, and produces more robust outcome (see figures 4,6).

4. We also adapt QAct to design a classifier which returns better calibrated probabilities. We show that, unlike the relevant, most recent baselines (RELU, DINOv2 (small)), QAct achieves constant calibration error across different severity of distortions.

## 1.2  Related Works

This work aims to address the failure mode described earlier. To the best of our knowledge, no existing literature directly addresses this issue. However, since the failure mode is widely persistent, different ideas have been explored to correct this.

*Related Works on Domain Generalization (DG):* The problem of domain generalization tries to answer the question – Can we use a classifier trained on one domain across several other related domains? The earliest known approach for this is *Transfer Learning* (Pan & Yang, 2010; Zhuang et al., 2021), where a classifier from a single domain is applied to a different domain with/without fine-tuning. Several approaches have been proposed to achieve DG, such as extracting domain-invariant features over single/multiple source domains (Ghifary et al., 2015; Akuzawa et al., 2019; Dou et al., 2019; Piratla et al., 2020; Hu et al., 2019), Meta Learning (Huang et al., 2020; Dou et al., 2019), Invariant Risk Minimization (Arjovsky et al., 2019). Self-supervised learning is another proposed approach which tries to extract features on large scale datasets in an unsupervised manner, the most recent among them being DINOv2 (Oquab et al., 2023) which is the current state-of-the-art[4]. Very large foundation models, such as GPT-4V, are also known to perform better with respect to distribution shifts (Han et al., 2023). Nevertheless, to the best of our knowledge, none of these models incorporates context distributions for classification.

*Related Works on Adaptive networks:* There have been several attempts to *adapt* the network to improve it's performance. Adaptive Batch Normalization and its variants (Li et al., 2017; Schneider et al., 2020; Zhou et al., 2024; Mirza et al., 2022) can be interpreted as special cases of context distribution modification, where only the mean and variance are adjusted. In contrast, methods that modify model parameters (Zhang et al., 2022) or alter the input distribution (Schwinn et al., 2022) preserve the pre-trained classifier but introduce a computationally expensive optimization step during inference. Empirical results indicate that adapting batch normalization statistics is at least as effective, if not superior, to these approaches. Furthermore, augmentation strategies (Hendrycks et al., 2020) are advantageous when domain-specific invariances are known, but such knowledge is not always available.

QAct differs from the aforementioned methods in that, while they focus on marginally improving adaptation by updating statistics, we instead model the entire context distribution. This approach is more general and yields superior performance, as demonstrated in Section 4.

## 1.3  Outline:

In section 2 we formally derive the quantile activation (QAct) and provide intuition, using a simple toy example, on why QAct adapts to distortions. Note that QAct diverges from existing train/test pipelines significantly. Section 3 discusses the practical implementation of QAct both at train and test time. This section also analyses the interoperability of QAct with standard layers and loss functions. Finally, in section 4 we evidence our claims by comparing QAct with other related methods discussed in section 1.2.

# 2  Quantile Activation

**Rethinking Outputs from a Neuron:** To recall – if $\mathbf{x}$ denotes the input, a typical neuron does the following – (i) Applies a linear transformation with parameters $w, b$, giving $w^t\mathbf{x} + b$ as the output, and (ii) applies a rectifier $g$, returning $g(w^t\mathbf{x} + b)$. Typically, $g$ is taken to be the ReLU activation - $g_{relu}(x) =$

---

[4] as per `https://paperswithcode.com/sota/domain-generalization-on-imagenet-c` accessed on 26 September 2024

$\max(0, x)$. Intuitively, we expect that each neuron captures an "abstract" feature, usually not understood by a human observer.

An alternate way to model a neuron is to consider it as predicting a latent variable y, where y = 1 if the feature is present and y = 0 if the feature is absent. Mathematically, we have the following model:

$$z = w^t \mathbf{x} + b + \epsilon \quad and \quad y = I[z \geq 0] \tag{2}$$

This is very similar to the standard latent variable model for logistic regression, with the main exception being, the *outputs* y *are not known* for each neuron beforehand. If y is known, it is rather easy to obtain the probabilities – $P(z \geq 0)$. Can we still predict the probabilities, even when y itself is a latent variable?

The authors in (Challa et al., 2024b) propose the following algorithm to estimate the probabilities:

1. Let $\{\boldsymbol{x}_i\}$ denote the set of input samples from the input distribution $\mathbf{x}$ and $\{z_i\}$ denote their corresponding latent outputs, which would be from the distribution z
2. Assign y = 1 whenever z > $(1 - \tau)^{th}$ quantile of z, and 0 otherwise. For a specific sample, we have $y_i = 1$ if $z_i > (1 - \tau)^{th}$ quantile of $\{z_i\}$
3. Fit the model $Q(x, \tau; \theta)$ to the dataset $\{((\boldsymbol{x}_i, \tau), y_i)\}$, and estimate the probability as,

$$P(y_i = 1) = \int_{\tau=0}^{1} I[Q(x, \tau; \theta) \geq 0.5]d\tau \tag{3}$$

**The key idea:** One can think of z above as pre-activations of a neuron. There are two key improvements which can be made – (i) The above procedure can be done at the level of each neuron and (ii) Instead of training a model with $\tau$ as input, we directly use quantiles of z for obtaining probabilities. Observe that in step 2., the labelling is done without resorting to actual ground-truth labels. This allows us to obtain the probabilities on the fly for any set of parameters, only by considering the quantiles of z. We detail these changes below.

**Defining the Quantile Activation QAct** Let z denote the pre-activation of the neuron, and let $\{z_i\}$ denote the samples from this distribution. Let $F_z$ denote the cumulative distribution function (CDF), and let $f_z$ denote the density of the distribution. Accordingly, we have that $F_z^{-1}(\tau)$ denotes the $\tau^{th}$ quantile of z. Using step (2) of the algorithm above, we define,

$$QAct(z) = \int_{\tau=0}^{1} I[z > F_z^{-1}(1 - \tau)]d\tau \overset{\text{Substitute}}{\underset{\tau \to (1-\tau)}{=}} \int_{\tau=0}^{1} I[z > F_z^{-1}(\tau)]d\tau \tag{4}$$

**Computing the gradient of QAct:** However, to use QAct in a neural network, we need to compute the gradient which is required for back-propagation. Let $\tau_z$ denote the quantile at which $F_z^{-1}(\tau_z) = z$. Then we have that $QAct(z) = \tau_z$ since $F_z^{-1}(\tau)$ is an increasing function. So, we have that $QAct(F_z^{-1}(\tau)) = \tau$. In other words, we have that $QAct(z)$ is $F_z(z)$, which is nothing but the CDF of z. Hence, we have,

$$\frac{\partial QAct(z)}{\partial z} = f_z(z) \tag{5}$$

where $f_z(z)$ denotes the density of the distribution.

**Grounding the Neurons:** With the above formulation, observe that since QAct is identical to CDF, it follows that, $QAct(z)$ is always a uniform distribution between 0 and 1, irrespective of the distribution z. When training numerous neurons in a layer, this could cause all the neurons to learn the same behaviour. Let $f$ denote a discriminatory feature which is to be learned but is present in only 25% of the population. However, the linear model which has to learn if $f$ is there (class 0) or not (class 1) should account for that imbalance. Gradient descent is known to fail here.

To correct this, we *enforce that positive values and negative values have equal weight.* Given the input distribution z, We perform the following transformation before applying QAct. Let

$$z^+ = \begin{cases} z & \text{if } z \geq 0 \\ 0 & \text{otherwise} \end{cases} \qquad z^- = \begin{cases} z & \text{if } z < 0 \\ 0 & \text{otherwise} \end{cases} \tag{8}$$

---

**Algorithm 1** Forward Propagation for a single neuron
___
**Input:** $[z_i]$ a vector of pre-activations, $0 < \tau_1 < \tau_2 < \cdots < \tau_{n_\tau} < 1$ - a list of quantile indices at which we compute the quantiles.

    Append two large values, $c$ and $-c$, to the vector $[z_i]$.

    Count $n_+$ = number of positive values, $n_-$ = number of negative values, and assign the weight $w_+ = 1/n_+$ to the positive values, and $w_- = 1/n_-$ to the negative values.

    Compute *weighted* quantiles $\{q_i\}$ at each of $\{\tau_i\}$ over the set $\{z_i\} \cup \{c, -c\}$

    Compute $QAct(z_i)$ using the function,

$$QAct(x) = \frac{1}{n_\tau} \sum_i I[x \geq q_i] \tag{6}$$

    Remember $[z_i]$, $w_+, w_-$, $[QAct(z_i)]$ for backward propagation.

    **return** $[QAct(z_i)]$
___

---

**Algorithm 2** Backward Propagation for a single neuron
___
**Input:** `grad_output`, $0 < \tau_1 < \tau_2 < \cdots < \tau_{n_\tau} < 1$ - a list of quantile indices at which we compute the quantiles.

**Context from Forward Propagation:** $[z_i]$, $w_+, w_-$, $[QAct(z_i)]$

    Obtain a weighted sample from $[z_i]$ with weights $w_+, w_-$ – (say) $S$.

    Obtain a kernel density estimate, using points from $S$, at each of the points in $z_i$ – (say) $\hat{f}_z(z_i)$

    Set,

$$\texttt{grad\_input} = \texttt{grad\_output} \odot [\hat{f}_z(z_i)] \tag{7}$$

    **return** `grad_input`
___

denote the truncated distributions. Then,

$$z^\ddagger = \begin{cases} z^+ & \text{with probability } 0.5 \\ z^- & \text{with probability } 0.5 \end{cases} \tag{9}$$

From definition of $z^\ddagger$, we get that the median of $z^\ddagger$ is 0. This grounds the input distribution to have the same positive and negative weight.

**Using pre-defined quantiles for computational efficiency:** In equation 6, one can theoretically consider $\sum_i I[x \geq x_i]$ instead of $\sum_i I[x \geq q_i]$. However the number of samples, $x_i$, can potentially be very large (order of $10^6$). Using uniform quantiles in $[0, 1]$ reduces the complexity when considering $n_\tau = 100$.

**Dealing with corner cases:** It is possible that during training, some neurons either only get positive values or only get negative values. However, for smooth outputs, one should still only give the weight of 0.5 for positive values. To handle this, we include two values $c$ (large positive) and $-c$ (large negative) for each neuron. Since, the quantiles are conventionally computed using linear interpolation, this allows the outputs to vary smoothly. We take $c = 100$ in this article.

**Estimating the Density for Back-Propagation:** Note that the gradient for the back propagation is given by the density of $z^\ddagger$ (weighted distribution). We use the *Kernel Density Estimation* (KDE), to estimate the density. We, (i) First sample $S$ points with weights $w_+, w_-$, and (ii) then estimate the density at all the input points $[z_i]$. This is point-wise multiplied with the backward gradient to get the gradient for the input. In this article we use $S = 1000$, which we observe gets reasonable estimates.

**Computational Complexity:** Computational Complexity (for a single neuron) is majorly decided by 2 functions – (i) Computing the quantiles has the complexity for a vector $[z_i]$ of size $n$ can be performed in

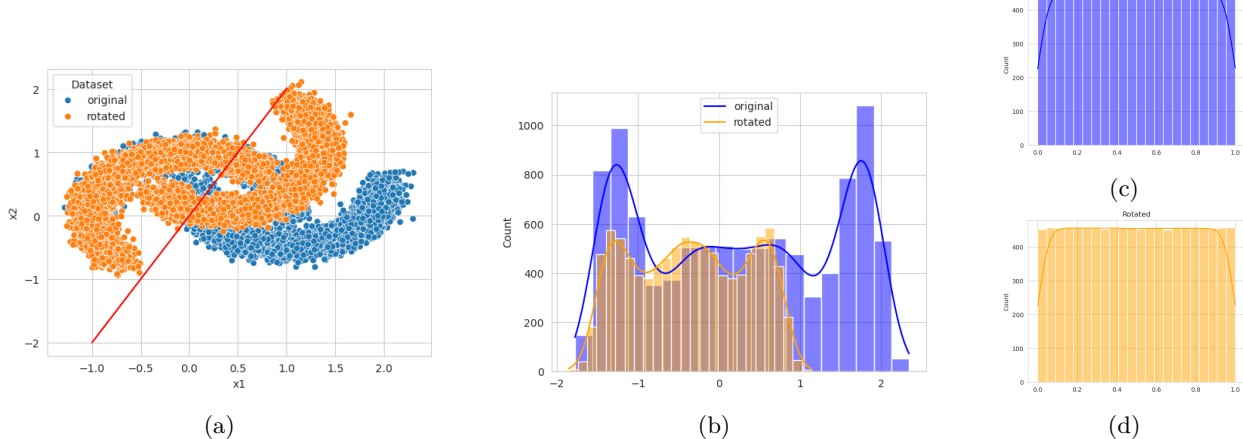

(a)                     (b)           (d)

Figure 3: Intuition behind quantile activation. (a) shows a simple toy distribution of points (blue), it's distortion (orange) and a simple line (red) on which the samples are projected to obtain activations. (b) shows the distribution of the pre-activations. (c) shows the distributions of the activations with QAct of the original distribution (blue). (d) shows the distributions of the activations with QAct under the distorted distribution (orange). Observe that the distributions match perfectly under small distortions. Note that even if the distribution matches perfectly, the quantile activation is actually a deterministic function.

$\mathcal{O}(n \log(n))$. Since this is log-linear in $n$, it does not increase the complexity drastically compared to other operations in a deep neural network. (ii) Computational complexity of the KDE estimates is $\mathcal{O}(Sn_\tau)$ where $S$ is the size of sample (weighted sample from $[z_i]$) and $n_\tau$ is the number of quantiles, giving a total of $\mathcal{O}(n \log(n) + Sn_\tau)$. In practice, we consider $S = 1000$ and $n_\tau = 100$ which works well, and hence does not increase with the batch size.

**Remark:** Algorithms 1, and 2 provide the pseudocode for the quantile activation. For stable training, in practice, we prepend and append the quantile activation with BatchNorm layers.

**Why QAct is robust to distortions?** To understand the idea behind quantile activation, consider a simple toy example in figure 3. For ease of visualization, assume that the input features (blue) are in 2 dimensions, and also assume that the line of the linear projection is given by the red line in figure 3a. Now, assume that the blue input features are rotated, leading to a different distribution (indicated here by orange). Since activations are essentially (unnormalized) signed distances from the line, we plot the histograms corresponding to the two distributions in figure 3b. As expected, these distributions are different. However, after performing the quantile activation in equation 4, we have that both are uniform distribution. This is illustrated in figures 3c and 3d. This behavior has a normalizing effect across different distributions, and hence has better distribution generalization than other activations.

**Explaining figure 1** Let $\{x_i, y_i\}$ denote the samples where $x_i \in \mathbb{R}^d$ ($d >> 2$) and $y_i \in \{0, 1\}$. Let $w^t x + b$ denote the classifier which separates the classes perfectly. The question we ask is – Under what transformations $x \to Ax$ does the classifier preserve it's accuracy?

As we discuss in appendix E, for standard linear models even if $A = c \times Id$ would not preserve the accuracy. On the other hand, for quantiles if $A^t w = \alpha w$ ($\alpha > 0$), then it preserves the accuracy. So, the class of matrices for which the accuracy is preserved is nothing but all *completions* of the matrix $A$ such that $A^t w = \alpha w$.

The network used in figure 1 projects the 2d points into high-dimensions (in a possibly non-linear way).The projection to high dimensions is such that the rotation in 2d preserves $A^t w = \alpha w$ in the high dimension for the final classification. Hence, the accuracy would be preserved. The important thing to note is – The

projection to the high dimensions, and the classifier in the subsequent layer, are all learned using SGD and none of these behaviors are hardcoded.

**How is QAct different from existing activation function?**  Standard activation functions such as ReLU take an input and apply a fixed function such as $\max(0, x)$. On the other hand, quantile activation can be thought of as a functional which takes as input both the value $x$ and also the cumulative distribution function $F$. In simple terms, we have

$$QAct(F, x) = F(x) \tag{10}$$

We refer to $F$ as a context distribution.

**How is QAct different from adaptive batch norm?**  Batch normalization is originally proposed to improve the gradient information during backpropagation. However, formally Batchnorm can also be thought of as a functional which takes in a distribution $X$ and the input $x$ and returns $(x - E[X])/Stdev(X)$. Thus, it only uses the first and second order information. Techniques as proposed in Li et al. (2017) update the distribution $X$ at test/inference time to allow for better genralization. However, batchnorm is *not invariant* to all monotonic transformations. This can easily be seen by considering the transform $x \to \exp(x)$. QAct on the other hand is invariant to all monotonic transforms.

## 3   Training with QAct

In the previous section, we described the procedure to adapt a single neuron to its context distribution. In this section we discuss how this extends to the Dense/Convolution layers, the loss functions to train the network and the inference aspect.

**Extending to standard layers:**  The extension of equation 4 to dense outputs is straightforward. A typical output of the dense layer would be of the shape $(B, N_c)$ - $B$ denotes the batch size, $N_c$ denotes the width of the network. The principle is - *The context distribution of a neuron is all the values which are obtained using the same parameters.* In this case, each of the values across the '$B$' dimension are considered to be samples from the context distribution.

For a convolution layer, the typical outputs are of the form - $(B, N_c, H, W)$ - $B$ denotes the size of the batch, $N_c$ denotes the number of channels, $H, W$ denotes the sizes of the images. In this case we should consider all values across the 1st,3rd and 4th dimension to be from the context distribution, since all these values are obtained using the same parameters. So, the number of samples would be $B \times H \times W$.

**Loss Functions:**  One can use any differentiable loss function to train with quantile activation. We specifically experiment with the standard Cross-Entropy Loss, Triplet Loss, and the recently proposed Watershed Loss in (Challa et al., 2024a) (see section 4). However, if one requires that the boundaries between classes adapt to the distribution, then learning similarities instead of boundaries can be beneficial. Both Triplet Loss and Watershed Loss fall into this category. We see that learning similarities does have slight benefits when considering the embedding quality.

**Inference with QAct:**  As stated before, we want to assign a label for classification based on the context of the sample. There exist two approaches for this – (1) One way is to keep track of the quantiles and the estimated densities for all neurons and use it for inference. This allows inference for a single sample in the traditional sense. However, this also implies that one would not be able to assign classes based on the context at evaluation. (2) Another way is to make sure that, even for inference on a single sample, we include several samples from the context distribution, but only use the output for a specific sample. This allows one to assign classes based on the context. In this article, we follow the latter approach.

**Importance of Batch size:**  A key assumption underlying QAct is – we assume the samples in the batch to be from the same distribution. However, this is not a strong restriction since there exists several techniques in the literature to assure that this assumption holds. In the extreme, one can even consider the batch size to be 1.

- In Zhang et al. (2022) the authors use additional augmentations at test time. Since one can be sure that these augmentations are from the same distribution as the original image, the implicit batch size can be increased.
- Another approach is to consider a *prior distribution* for each sample and update it using pre-activations from the given sample by bootstrapping. This is equivalent to the approach considered in Schneider et al. (2020).

Optimal performance is attained when the inference distribution exhibits sufficient diversity. In this work, we assume this condition holds, as it is satisfied by our experimental datasets.

**Quantile Classifier:** Observe that the proposed QAct (without normalization) returns the values in $[0, 1]$ which can be interpreted as probabilities. Hence, one can also use this for the classification layer. Nonetheless, two changes are required – (i) Traditional softmax used in conjunction with negative-log-likelihood loss already considers "relative" activations of the classification in normalization. However, QAct does not. Hence, one should use Binary-Cross-Entropy loss with QAct, which amounts to one-vs-rest classification. (ii) Also, unlike a neuron in the middle layers, the bias of the neuron in the classification layer depends on the class imbalance. For instance, with 10 classes, one would have only $1/10$ of the samples labelled 1 and $9/10$ of the samples labelled 0. To address this, we require that the median of the outputs be at 0.9, and hence weight the positive class with 0.9 and the negative class with 0.1 respectively. In this article, whenever QAct is used, we use this approach for inference.

We observe that (figures 14 and 15) using quantile classifier on the learned features in general improves the consistency of the calibration error and also leads to the reducing the calibration error. In this article, for all networks trained with quantile activation, we use quantile classifier to compute the accuracy/calibration errors.

## 4 Evaluation

To summarize, we make the following changes to the existing classification pipeline – (i) Replace the usual ReLU activation with QAct and (ii) Use triplet or watershed loss instead of standard cross-entropy loss. We expect this framework to learn context dependent features, and hence be robust to distortions. (iii) Also, use quantile classifier to train the classifier on the embedding for better calibrated probabilities.

**Evaluation Protocol:** To evaluate our approach, we consider the datasets developed for this purpose – CIFAR10C, CIFAR100C, TinyImagenetC (Hendrycks & Dietterich, 2019), MNISTC (Mu & Gilmer, 2019). These datasets have a set of 15 distortions at 5 severity levels. To ensure diversity we evaluate our method on 4 architectures – (overparametrized) LeNet, ResNet18 (He et al., 2016) (11M parameters), VGG (Simonyan & Zisserman, 2015)(15M parameters) and DenseNet (Huang et al., 2017) (1M parameters). The code to reproduce the results can be found at `https://github.com/adityac20/quantile_activation`.

**Baselines for Comparison:** To our knowledge, there exists no other framework which proposes classification based on context distribution. So, for comparison, we consider standard ReLU activation (Fukushima, 1970), pReLU (He et al., 2015), and SELU (Klambauer et al., 2017) for all the architectures stated above. Also, we compare our results with DINOv2 (small) (Oquab et al., 2023) (22M parameters) which is current state-of-the-art for domain generalization. Note that for DINOv2, architecture and datasets used for training are substantially different (and substantially larger) from what we consider in this article. Nevertheless, we include the results for understanding where our proposed approach lies on the spectrum. We consider the small version of DINOv2 to match the number of parameters with the compared models. We also compare QAct with other adaptive algorithms (Schneider et al., 2020; Mirza et al., 2022; Zhang et al., 2022; Sun et al., 2020; Zhou et al., 2024) on CIFAR10C.

**Metrics:** We consider the following metrics – (i)Accuracy (ACC), (ii) calibration error (ECE) (Kumar et al., 2019) (both marginal and Top-Label) (iii) mean average precision at $K$ (MAP@K) to evaluate the embedding, (iv) Drop in accuracy (ACC_DROP) which measures the difference in accuracy at distortion $i$ ($i = 1 \ldots 5$) and at distortion 0 (standard test set). For the case of ReLU/pReLU/SELU activation with

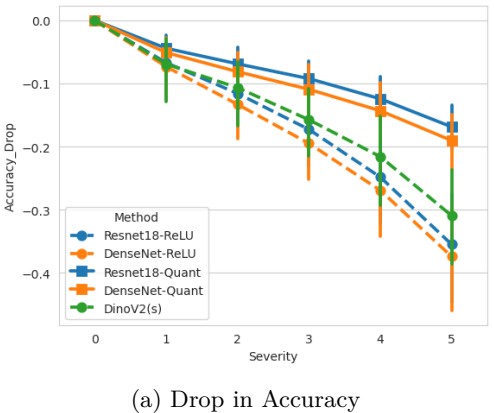
(a) Drop in Accuracy

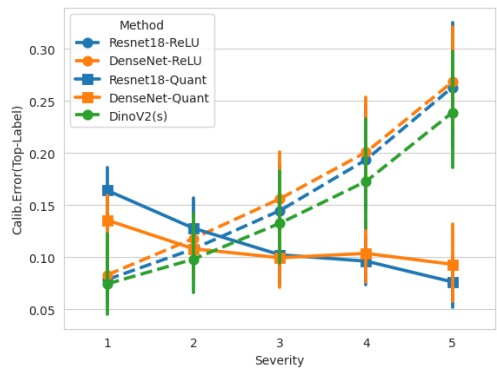
(b) Top-Label Calibration Error

Figure 4: Comparing QAct with ReLU activation and DINOv2 (small) on CIFAR10C. We observe that, while at low severity of distortions QAct has a similar accuracy as existing pipelines, at higher levels the drop in accuracy is substantially smaller than existing approaches. With respect to calibration, we observe that the calibration error remains constant (up to standard deviations) across distortions.

Cross-Entropy, we use the logistic regression trained on the train set embeddings, and for QAct we use the calibrated linear classifier, as proposed above. We do not perform any additional calibration and use the probabilities. We discuss a selected set of results in the main article. Please see appendix C for more comprehensive results.

Calibration error measures the reliability of predicted probabilities. In simple words, if one predicts 100 samples with (say) probability 0.7, then we expect 70 of the samples to belong to class 1 and the rest to class 0. This is measured using either the marginal or top-label calibration error. We refer the reader to (Kumar et al., 2019) for details, which also provides an implementation to estimate the calibration error.

**Remark:** For all the baselines we use the standard Cross-Entropy loss for training. For inference on corrupted datasets, we retrain the last layer with logistic regression on the train embedding and evaluate it on test/corrupted embedding. For QAct, we as a convention use watershed loss unless otherwise stated, for training. For inference, we train the Quantile Classifier on the train embedding and evaluate it on test/corrupted embedding.

**The proposed QAct approach is robust to distortions:** In fig. 4 we compare the proposed QAct approach with predominant existing pipeline – ReLU+Cross-Entropy and DINOv2(small) on CIFAR10C. In figure 4a we see that as the severity of the distortion increases, the accuracy of ReLU and DINOv2 drops significantly. On the other hand, while at small distortions the results are comparable, as severity increases QAct performs substantially better than conventional approaches. At severity 5, QAct outperforms DINOv2. On the other hand, we observe that in figure 4b, the calibration error stays consistent across distortions.

**How much does QAct depend on the loss function?** Figure 5a compares the watershed classifier with other popular losses – Triplet and Cross-Entropy. We see that all the loss functions perform comparably when used in conjunction with QAct. We observe that watershed has a slight improvement when considering MAP and hence, we consider that as the default setting. However, we point out that QAct is compatible with several loss functions as well.

**QAct vs ReLU/pReLU/SELU activations:** To verify that most existing activations do not share the robustness property of QAct, we compare QAct with other activations in figures 5b and 5c. We observe that QAct is greatly more robust with respect to distortions in both accuracy and calibration error than other activation functions.

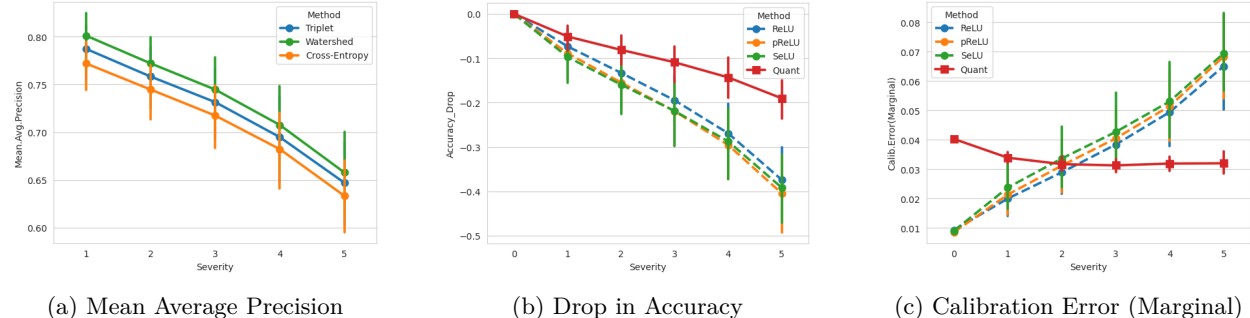

(a) Mean Average Precision      (b) Drop in Accuracy      (c) Calibration Error (Marginal)

Figure 5: (a) Dependence on Loss functions. Here we compare watershed with other popular loss functions – Triplet and Cross-Entropy when used with QAct. We see that watershed performs slightly better with respect to MAP. (b) Comparing QAct with other popular activations – ReLU/pReLU/SELU with respect to drop in accuracy. (c) Comparing QAct with other popular activations – ReLU/pReLU/SELU with respect to Calibration Error (Marginal). From both (b) and (c) we can conclude that QAct is notably more robust across distortions than several of the existing activation. All the plots use ResNet18 with CIFAR10C dataset.

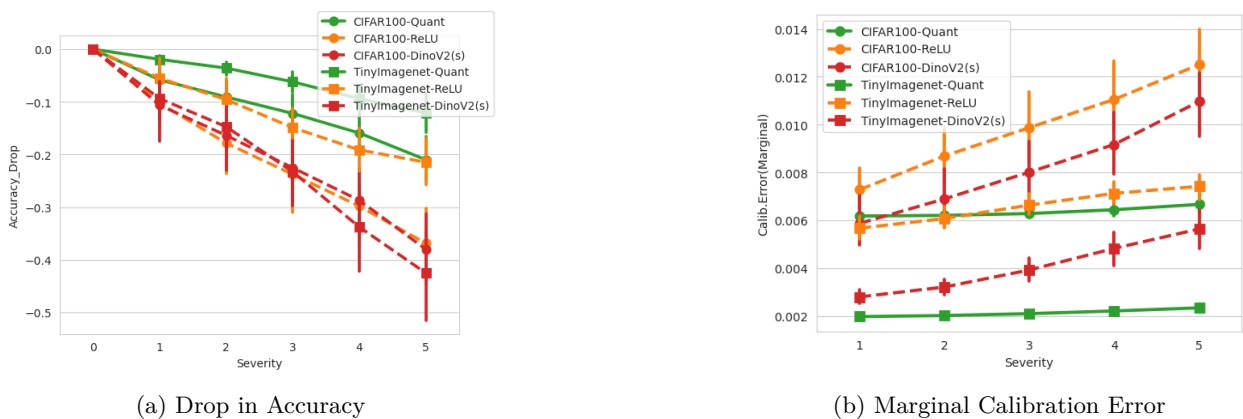

(a) Drop in Accuracy            (b) Marginal Calibration Error

Figure 6: Results on CIFAR100C/TinyImagenetC. We compare QAct+watershed to ReLU and DinoV2 small on CIFAR100C/TinyImagenetC dataset with ResNet18. Note that the observations are consistent with CIFAR10C. (a) shows drop in accuracy across distortions. Observe that QAct gets much smaller drop in accuracy than DINOv2(s) across all distortions, even if DINOv2 has 22M parameters as compared to Resnet18 11M parameters and is trained on larger datasets. (b) shows how calibration error (marginal) changes across severities. While other approaches lead to an increase in calibration error, QAct has similar calibration error across distortions.

| Model | Severity →
Method ↓ | 0→1 | 0→2 | 0→3 | 0→4 | 0→5 |
|---|---|---|---|---|---|---|
| Resnet18 (11M) | QAct | **0.0442** | **0.0688** | **0.0922** | **0.1242** | **0.1686** |
| Resnet26 (16M) | Batch Norm (Schneider et al., 2020) | 0.1280 | 0.1580 | 0.1850 | 0.2190 | 0.2600 |
| | DUA (Mirza et al., 2022) | 0.0670 | 0.0930 | 0.1180 | 0.1500 | 0.1990 |
| | MEMO (Zhang et al., 2022) | 0.0483 | 0.0765 | 0.1117 | 0.1543 | 0.2239 |
| | TTT (Sun et al., 2020) | 0.0710 | 0.1030 | 0.1360 | 0.1780 | 0.2450 |
| WRN-40-4 (9M) | AFN(Zhou et al., 2024) | 0.0585 | 0.0855 | 0.1145 | 0.1525 | 0.1815 |
| | ASRNorm(Zhou et al., 2024) | 0.0460 | 0.0790 | 0.1110 | 0.1540 | 0.2110 |

Table 1: Comparing QAct with other adaptive approaches on CIFAR10C. We measure the drop in accuracy with respect to uncorrupted dataset.

**Results on Larger Datasets:** To verify that our observations hold for larger datasets, we use CIFAR100C/TinyImagenetC to compare the proposed QAct+watershed with existing approaches. We observe on figure 6 that QAct performs comparably well as DINOv2, although DINOv2(s) has 22M parameters and is trained on significantly larger datasets. Moreover, we also observe that QAct has approximately constant calibration error across distortions, as opposed to a significantly increasing calibration error for ReLU or DINOv2.

**Comparison with other adaptive algorithms:** As stated before, there have been other attempts to *adapt* the neural network to the distribution. Table 1 compares the quantile activation with these approaches on CIFAR10C. Since, the methods are diverse, we compare the approached using the drop in accuracy. We see that QAct outperforms all the competing approaches. This can be attributed to the properties of QAct – Normalizing effect as discussed in section 2 and that quantiles are *invariant* to monotonic transformations.

## 5 Conclusion And Future Work

To summarize, traditional classification systems do not consider the "context distributions" when assigning labels. In this article, we propose a framework to achieve this by – (i) Making the activation adaptive by using quantiles and (ii) Learning a kernel instead of the boundary for the last layer. We show that our method is more robust to distortions by considering MNISTC, CIFAR10C, CIFAR100C, TinyImagenetC datasets across varying architectures.

The scope of this article is to provide a proof of concept and a framework for performing inference in a context-dependent manner. We outline several potential directions for future research:

I. The key idea in our proposed approach is that the quantiles capture the distribution of each neuron from the batch of samples, providing outputs accordingly. This poses a challenge for large datasets, and we have discussed two potential solutions: (i) remember the quantiles and density estimates for single sample evaluation, or (ii) ensure that a batch of samples from the same distribution is processed together. We adopt the latter method in this article. An alternative approach would be to *learn the distribution of each neuron* using auxiliary loss functions, adjusting these distributions to fit the domain at test time. This gives us more control over the network at test time compared to current workflows. If the networks are very large, where batch sizes cannot be big – there exists several strategies such as checkpointing to implicitly increase the batch size.

II. Since the aim of the article was to establish a proof-of-concept, we did not focus on scaling, and use only a single GPU for all the experiments. To extend it to multi-GPU training, one needs to synchronize the quantiles across GPU, in a similar manner as that for Batch-Normalization. We expect this to improve the statistics, and to allow considerably larger batches of training.

III. On the theoretical side, there is an interesting analogy between our quantile activation and how a biological neuron behaves. It is known that when the inputs to a biological neuron change, the neuron adapts to these changes (Clifford et al., 2007). Quantile activation does something very

similar, which leads to an open question – can we establish a formal link between the adaptability of a biological neuron and the accuracy of classification systems?

IV. Another theoretical direction to explore involves considering distributions not just at the neuron level, but at the layer level, introducing a high-dimensional aspect to the problem. The main challenge here is defining and utilizing *high dimensional quantiles*, which remains an open question (Koenker, 2005).

**Acknowledgments**

Aditya Challa and Laurent Najman acknowledges the support from CEFIPRA(68T05-1) . Snehanshu Saha, Aditya Challa, Sravan Danda would like to thank the Anuradha and Prashanth Palakurthi Center for Artificial Intelligence Research (APPCAIR) and ANRF CRG-DST (CRG/2023/003210) for support. Snehanshu Saha acknowledges SERB SURE-DST (SUR/2022/001965) and the DBT-Builder project (BT/INF/22/SP42543/2021), Govt. of India for partial support. Sravan Danda would like to thank ANRF for funding from SERB-SURE project number SUR/2022/002735.

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

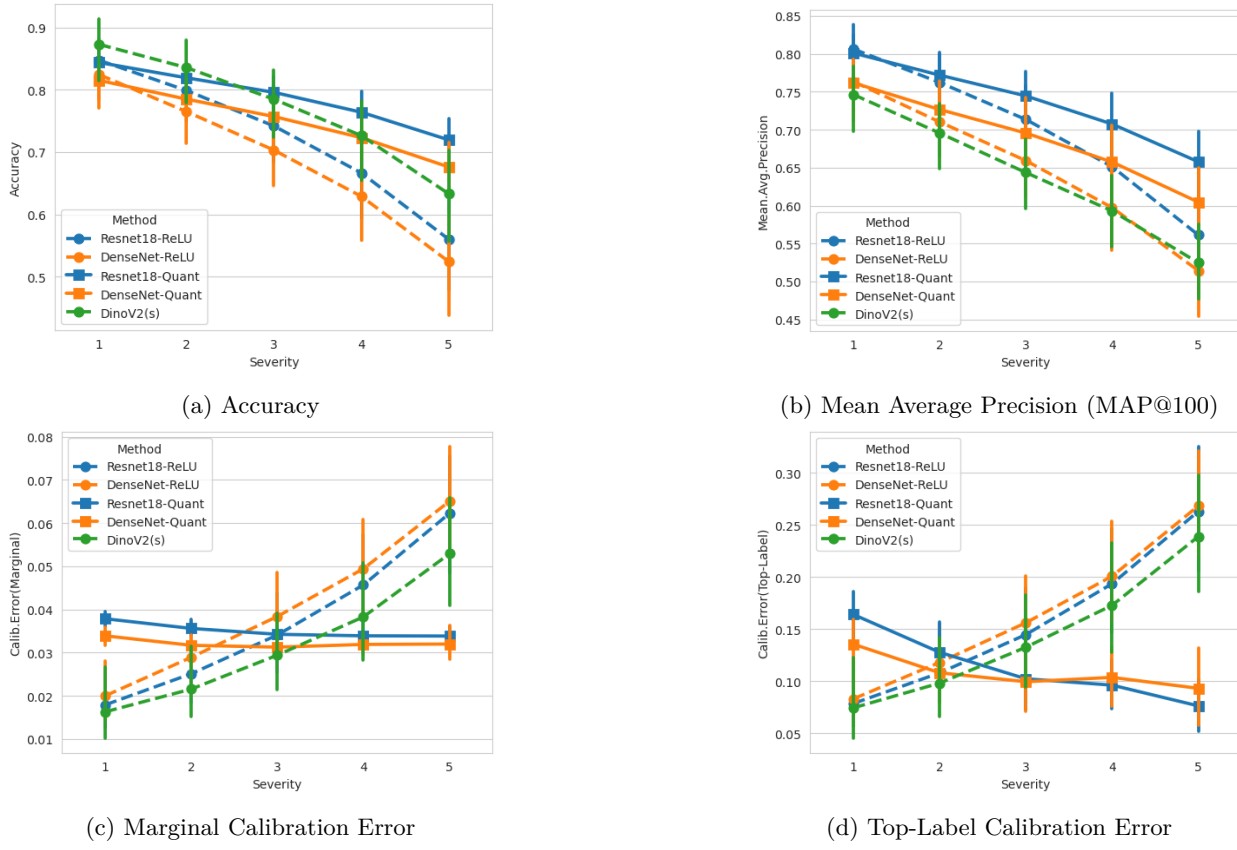

(a) Accuracy

(b) Mean Average Precision (MAP@100)

(c) Marginal Calibration Error

(d) Top-Label Calibration Error

Figure 7: Comparing QAct with ReLU activation and DINOv2 (small) on CIFAR10C

Fuzhen Zhuang, Zhiyuan Qi, Keyu Duan, Dongbo Xi, Yongchun Zhu, Hengshu Zhu, Hui Xiong, and Qing He. A comprehensive survey on transfer learning. *Proc. IEEE*, 2021.

## A    Experiment details for figure 2

We consider the features obtained from ResNet18 with both QAct and ReLU activations for the datasets of CIFAR10C with gaussian_noise at all the severity levels. Hence, we have 6 datasets in total. To use TSNE for visualization, we consider 1000 samples from each dataset and obtain the combined TSNE visualizations. Each figure shows a scatter plot of the 2d visualization for the corresponding dataset.

## B    Compute Resources and Other Experimental Details

All experiments were performed on a single NVidia GPU with 32GB memory with Intel Xeon CPU (10 cores). For training, we perform an 80:20 split of the train dataset with seed 42 for reproducibility. All networks are initialized using default pytorch initialization technique.

We use Adam optimizer with initial learning rate $1e-3$. We use ReduceLRonPlateau learning rate scheduler with parameters – factor=0.1, patience=50, cooldown=10, threshold=0.01, threshold_mode=abs, min_lr=1e-6. We monitor the validation accuracy for learning rate scheduling. We also use early_stopping when the validation accuracy does not increase by 0.001.

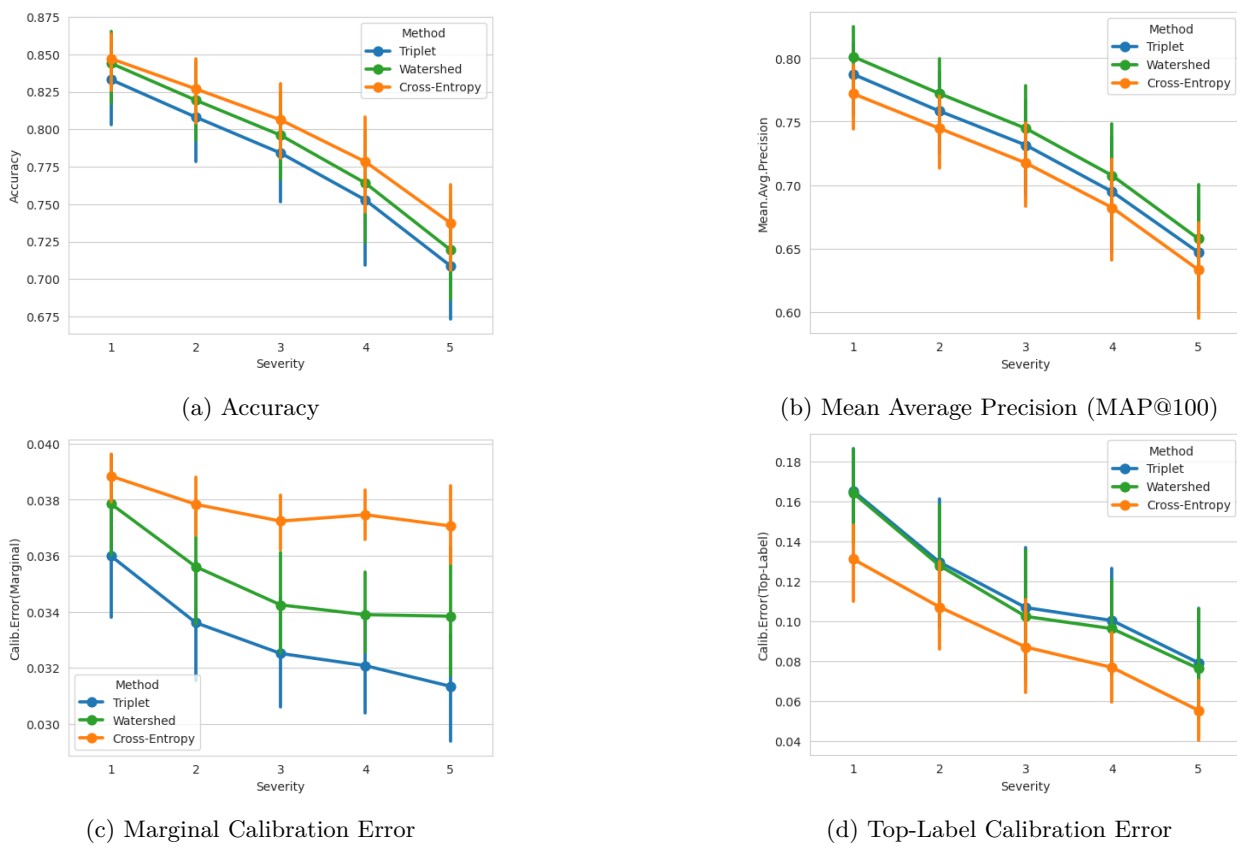

(a) Accuracy

(b) Mean Average Precision (MAP@100)

(c) Marginal Calibration Error

(d) Top-Label Calibration Error

Figure 8: Triplet vs Watershed vs Cross-Entropy using Resnet18+CIFAR10C

## C  Extended Results Section

**Comparing QAct + watershed and ReLU+Cross-Entropy:**  Figure 7 shows the corresponding results. The first experiment compares QAct + watershed with ReLU + Cross-Entropy on two standard networks – ResNet18 and DenseNet. With respect to accuracy, we observe that while at severity 0, ReLU + Cross-Entropy slightly outperforms QAct + watershed, as severity increases QAct + watershed is far more stable. We even outperform DINOv2(small) (22M parameters) at severity 5. Moreover, with respect to calibration error, we see a consistent trend across distortions. As (Challa et al., 2024b) argues, this helps in building more robust systems compared to one where calibration error increases across distortions.

**Remark:**

**Does loss function make a lot of difference?**  Figure 8 compares three different loss functions Watershed, Triplet and Cross-Entropy when used in conjunction with QAct. We observe similar trends across all loss functions. However, Watershed performs better with respect to Mean Average Precision (MAP) and hence we use this as a default strategy.

Why Mean-Average-Precision? – We argue that the key indicator of distortion invariance should be the quality of embedding. While, accuracy (as measured by a linear classifier) is a good metric, a better one would be to measure the Mean-Average-Precision. With respect to calibration error, due to the scale on the Y-axis, the figures suggest reducing calibration error. However, the standard deviations overlap, and hence, these are assumed to be constant across distortions.

**How well does watershed perform when used with ReLU activation?**  Figure 9 shows the corresponding results. We observe that both the watershed loss and cross-entropy have large overlaps in the

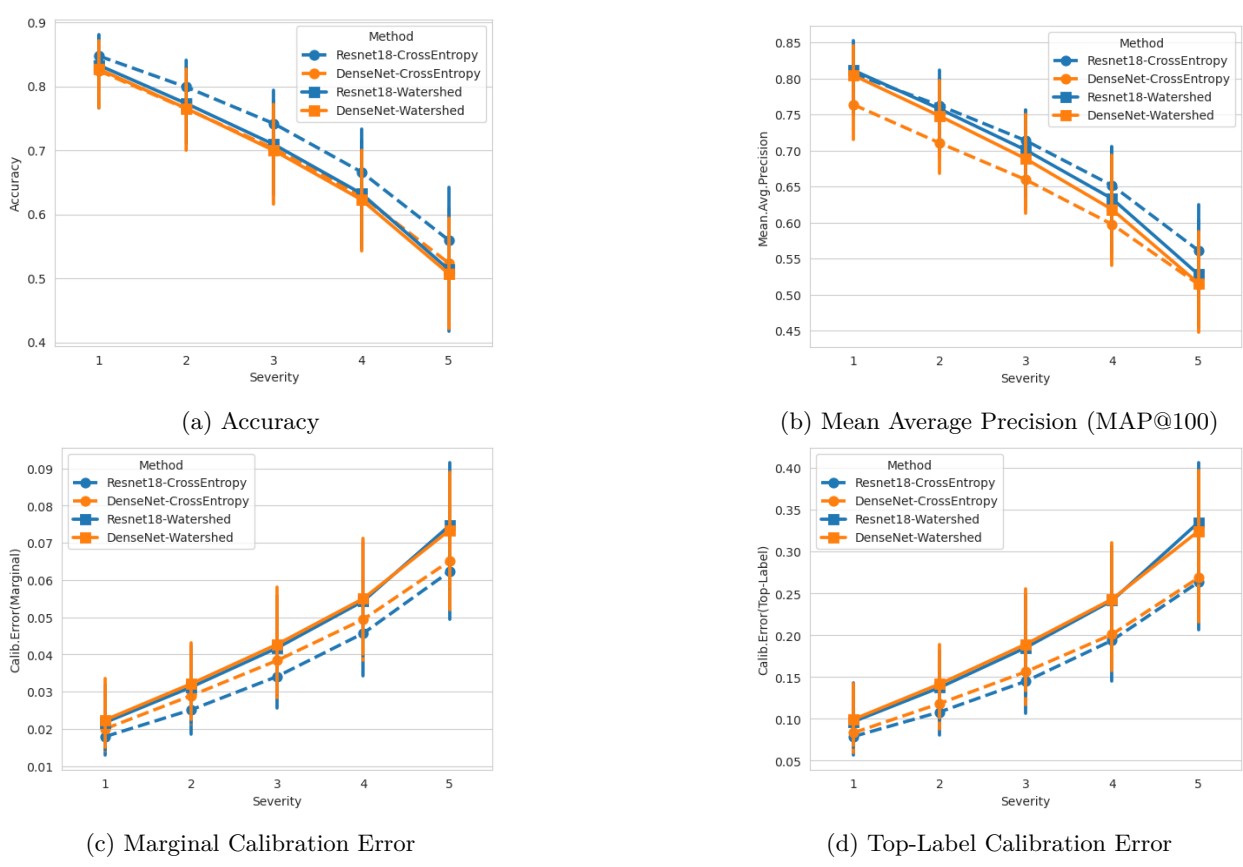

(a) Accuracy

(b) Mean Average Precision (MAP@100)

(c) Marginal Calibration Error

(d) Top-Label Calibration Error

Figure 9: Watershed vs Cross-Entropy when using ReLU activation using CIFAR10C

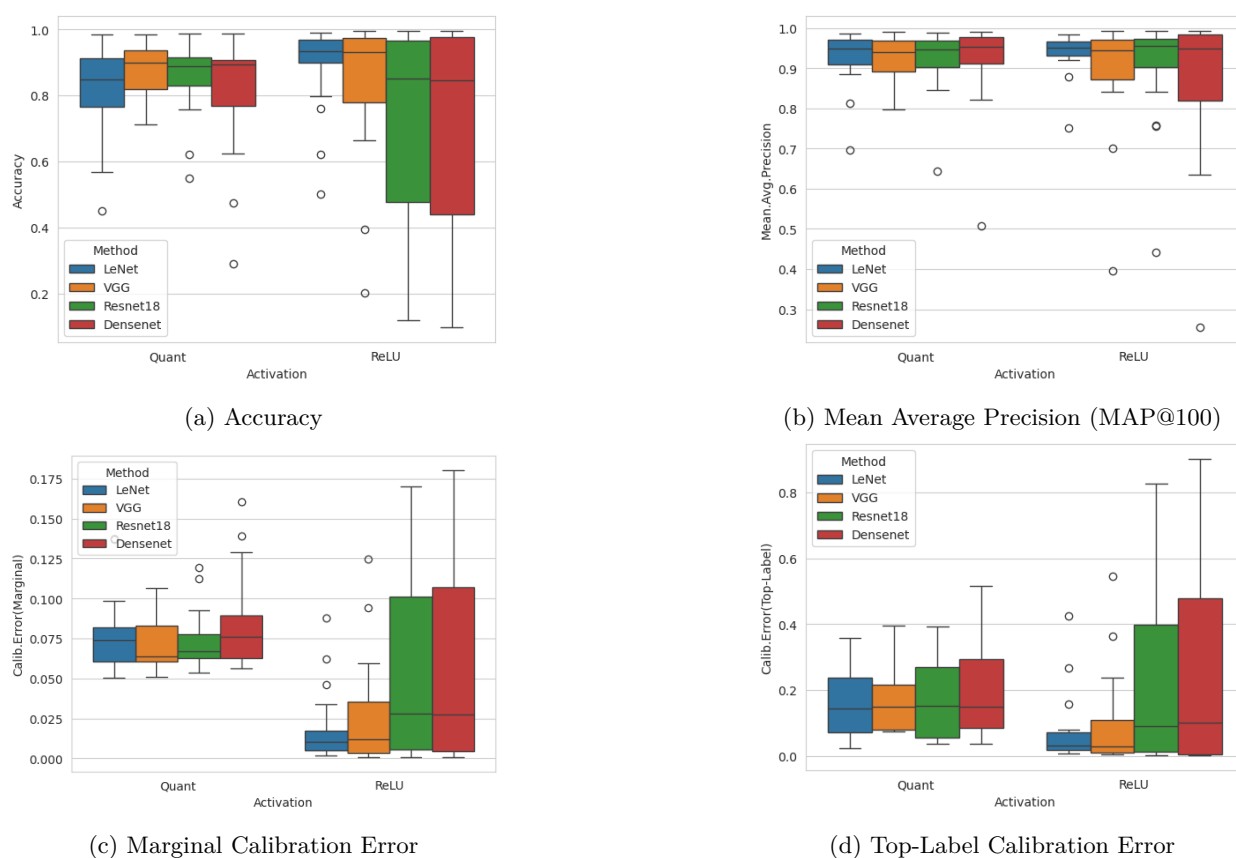

(a) Accuracy

(b) Mean Average Precision (MAP@100)

(c) Marginal Calibration Error

(d) Top-Label Calibration Error

Figure 10: Results on MNIST

standard deviations at all severity levels. So, this shows that, when used in conjunction with ReLU watershed and cross-entropy loss are very similar. But in conjunction with QAct, we see that watershed has a slightly higher Mean-Average-Precision.

**What if we consider an easy classification task?** In figure 10, we perform the comparison of QAct+Watershed and ReLU and cross-entropy on MNISTC dataset. Across different architectures, we observe a lot less variation (standard deviation) of QAct+Watershed compared to ReLU and cross-entropy. This again suggests robustness against distortions of QAct+Watershed.

**Comparing with other popular activations:** Figures 11 and 12 shows the comparison of QAct with ReLU, pReLU and SeLU. We observe the same trend across ReLU, pReLU and SeLU, while QAct is far more stable across distortions.

**Results on CIFAR100/TinyImagenetC:** Figure 13 compares QAct+Watershed and ReLU+Cross-Entropy on CIFAR100C dataset. We also include the results of QAct+Cross-Entropy vs. ReLU+Cross-Entropy on TinyImagenetC. The results are consistent with what we observe on CIFAR10C, and hence, draw the same conclusions as before.

**Effect of Quantile Classifier:** Figures 14 and 15 shows the effect of quantile classifier on standard ResNet10/DinoV2 outputs with CIFAR10C/CIFAR100C datasets. While the accuracy values are almost equivalent, we observe a "flatter" trend of the calibration errors, sometimes reducing the error as in the case of CIFAR100C.

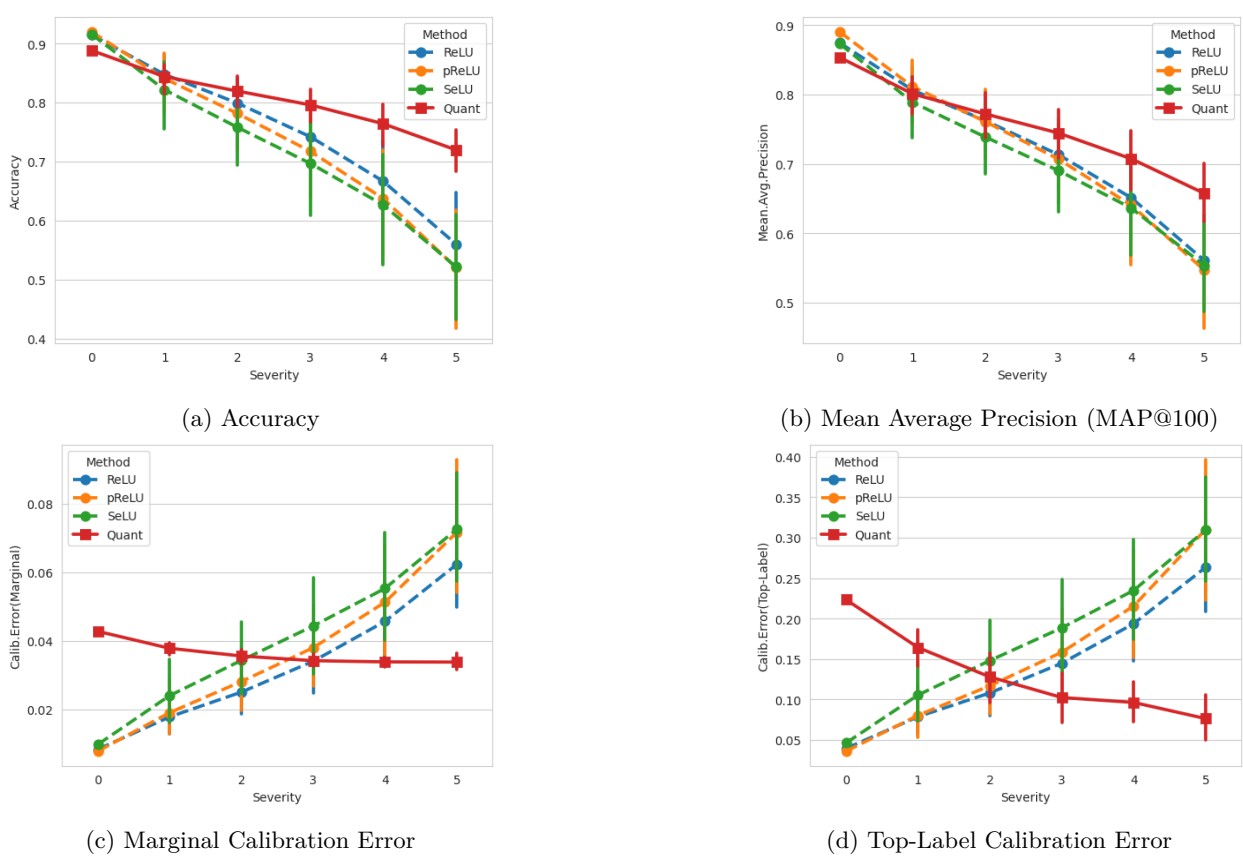

(a) Accuracy

(b) Mean Average Precision (MAP@100)

(c) Marginal Calibration Error

(d) Top-Label Calibration Error

Figure 11: QActvs ReLU vs pReLU vs Selu activations on ResNet18 + CIFAR10C

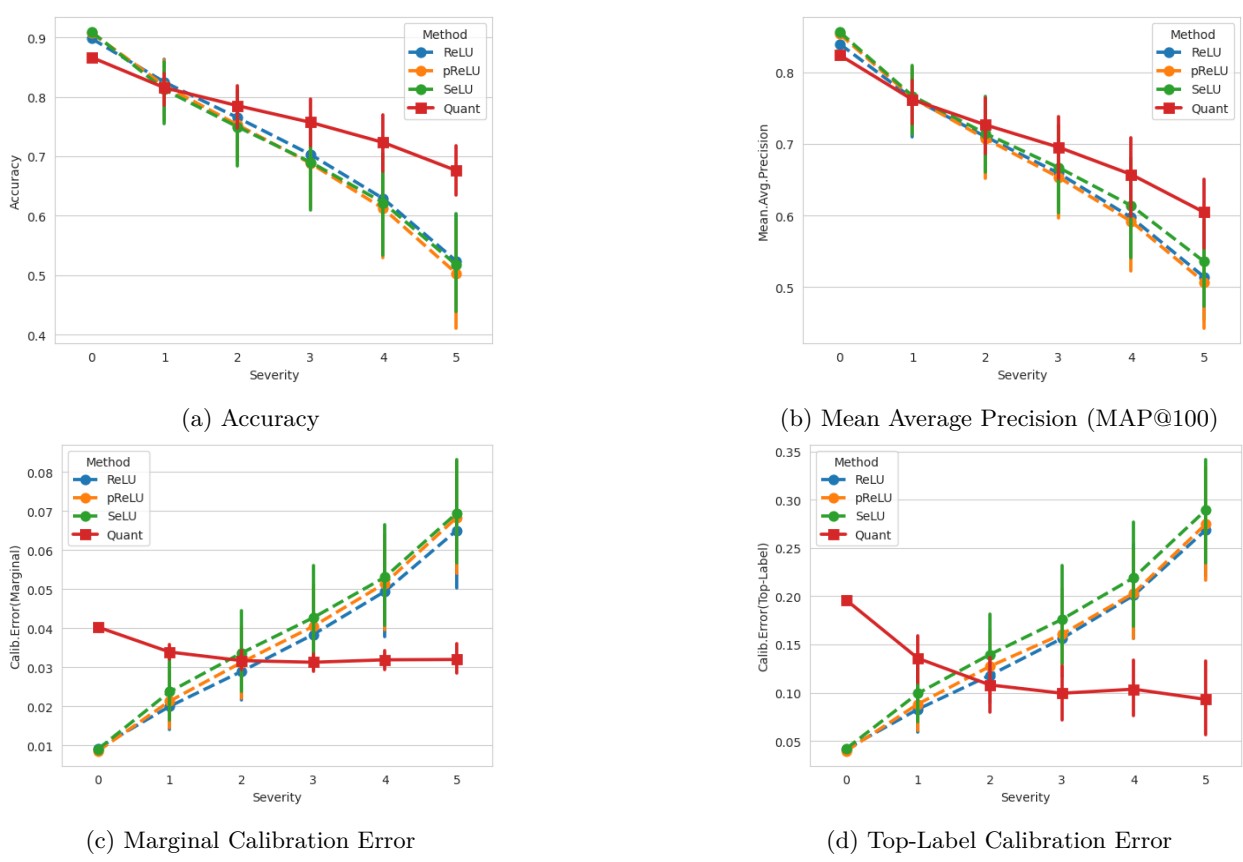

(a) Accuracy

(b) Mean Average Precision (MAP@100)

(c) Marginal Calibration Error

(d) Top-Label Calibration Error

Figure 12: QActvs ReLU vs pReLU vs Selu activations on Densenet+CIFAR10C

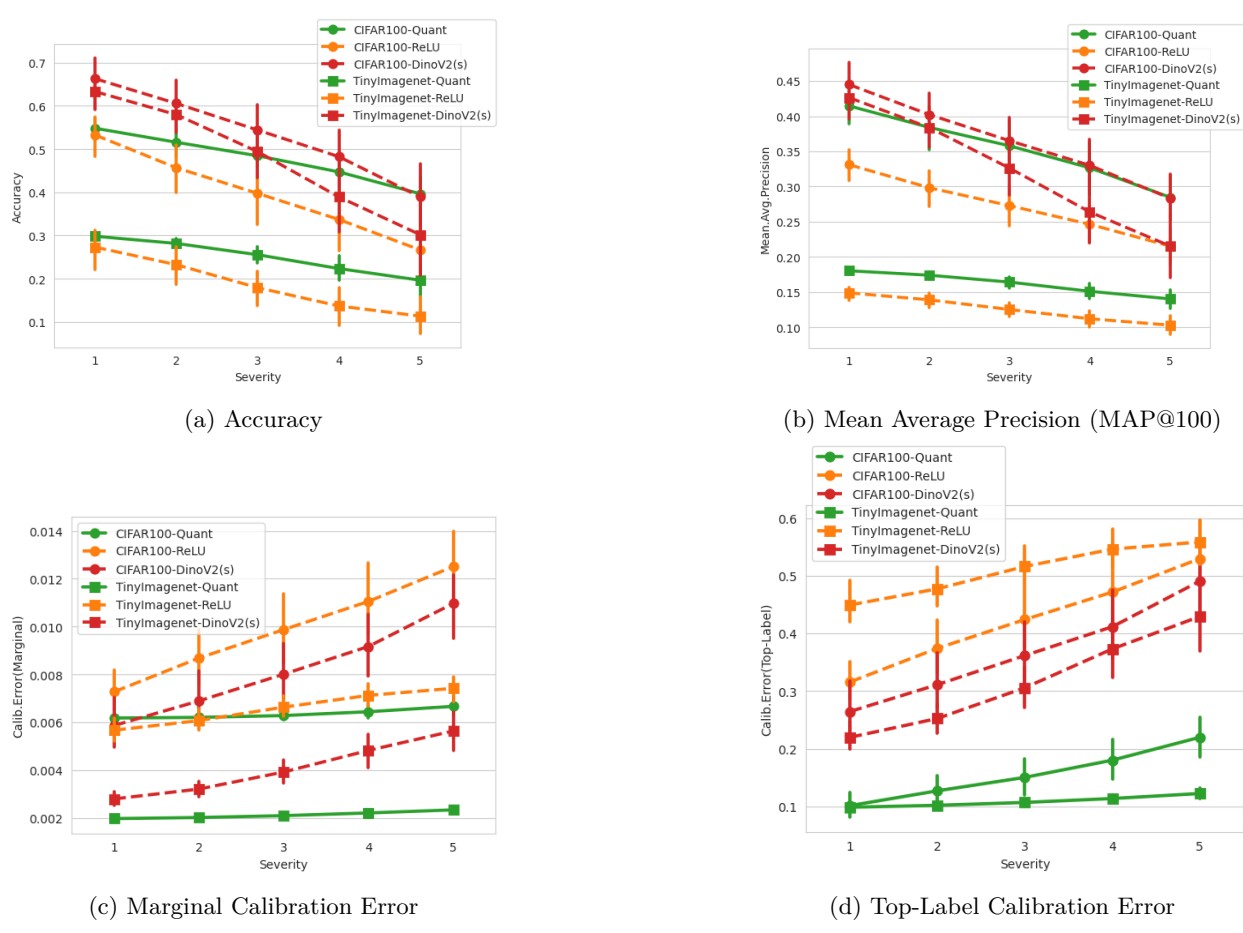

(a) Accuracy

(b) Mean Average Precision (MAP@100)

(c) Marginal Calibration Error

(d) Top-Label Calibration Error

Figure 13: QActvs ReLU on Resnet18+CIFAR100C/TinyImagenetC

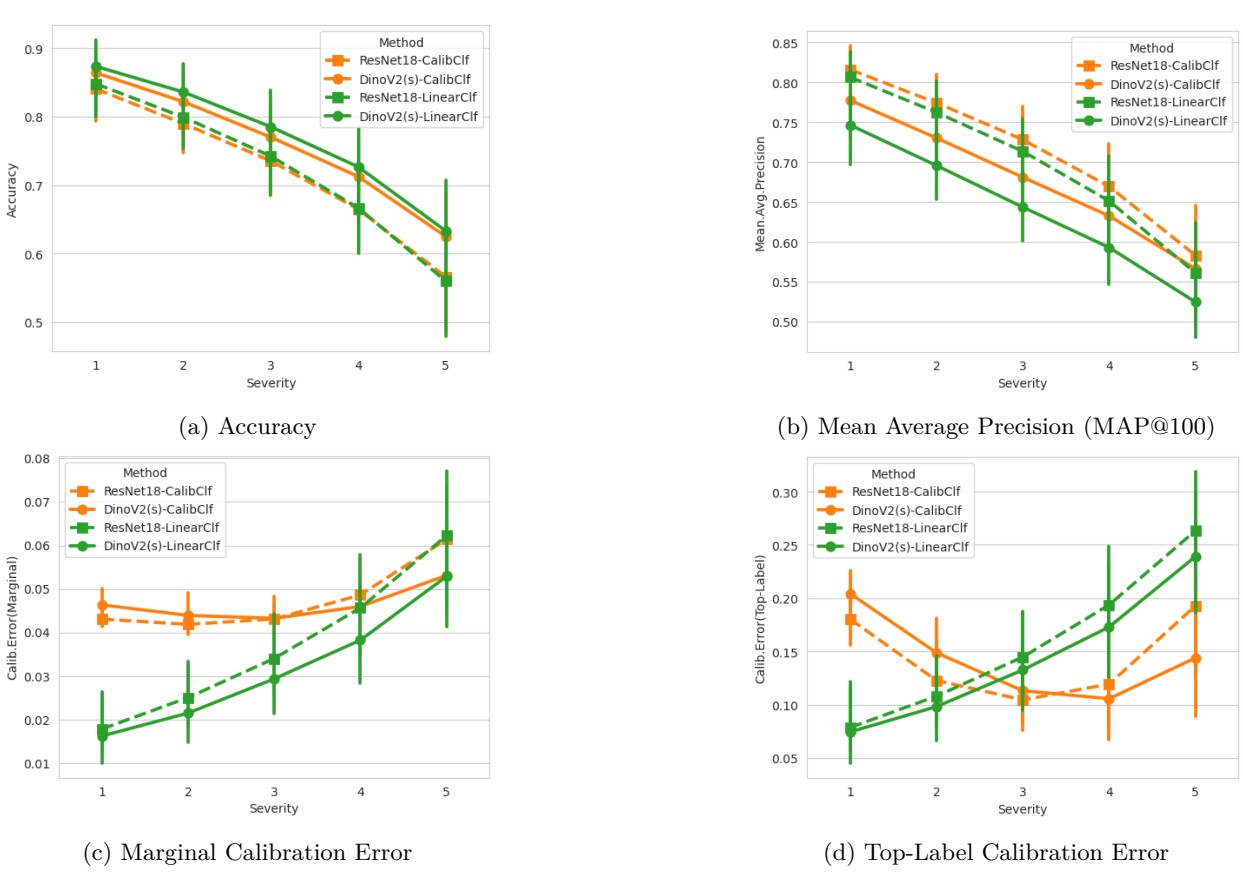

(a) Accuracy

(b) Mean Average Precision (MAP@100)

(c) Marginal Calibration Error

(d) Top-Label Calibration Error

Figure 14: Effect of Quantile Classifier. We use ResNet18 and DinoV2 architectures on CIFAR10C.

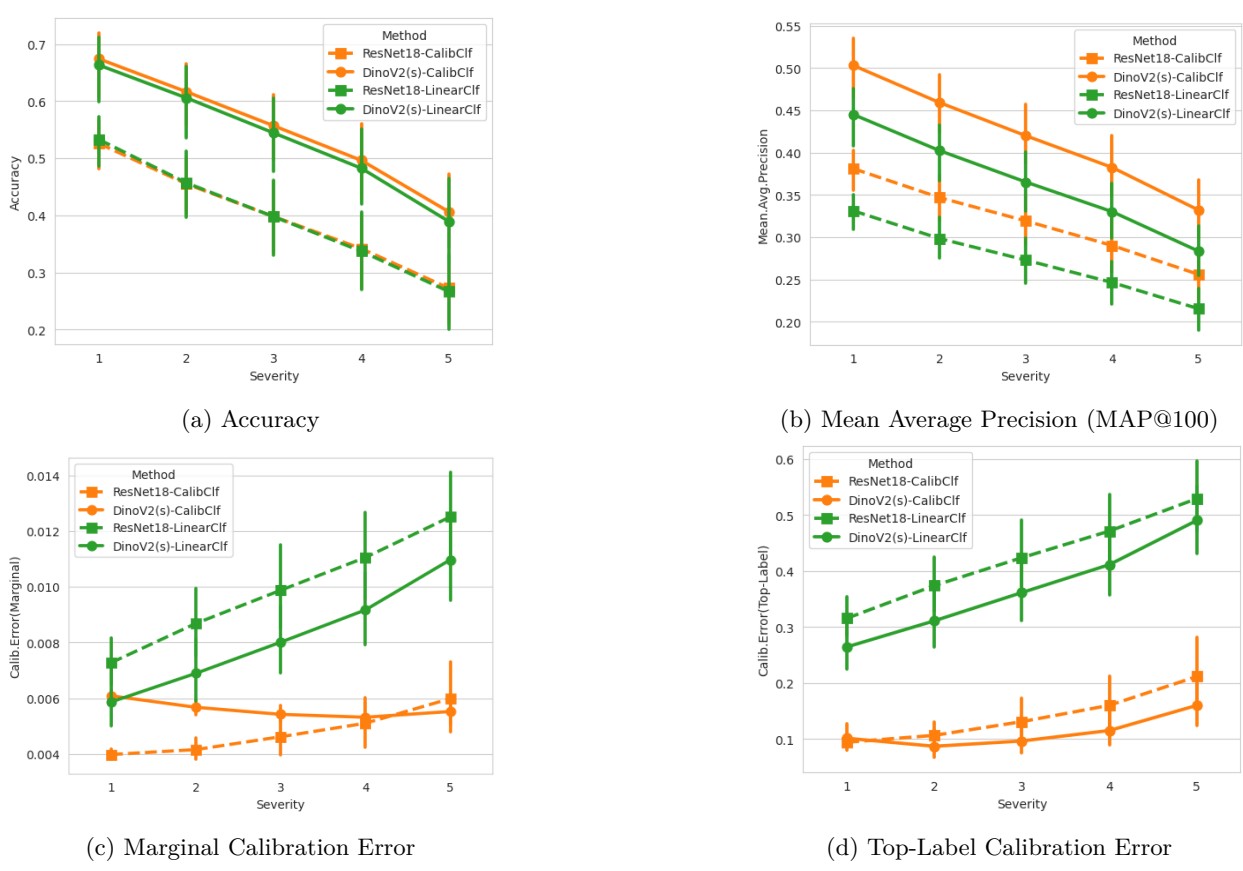

(a) Accuracy

(b) Mean Average Precision (MAP@100)

(c) Marginal Calibration Error

(d) Top-Label Calibration Error

Figure 15: Effect of Quantile Classifier. We use ResNet18 and DinoV2 architectures on CIFAR100C.

| | | 0→1 | 0→2 | 0→3 | 0→4 | 0→5 | 1→2 | 1→3 | 1→4 | 1→5 |
|---|---|---|---|---|---|---|---|---|---|---|
| Dataset | Model/Method | | | | | | | | | |
| CIFAR10 | Resnet18-ReLU | 6.70 | 11.57 | 17.27 | 24.82 | 35.49 | 4.87 | 10.57 | 18.12 | 28.79 |
| | DinoV2(s) | 6.92 | 10.66 | 15.74 | 21.58 | 30.93 | 3.74 | 8.82 | 14.67 | 24.02 |
| | Resnet18-Quant | 4.42 | 6.88 | 9.22 | 12.42 | 16.86 | 2.46 | 4.80 | 8.00 | 12.44 |
| CIFAR100 | Resnet18-ReLU | 10.25 | 17.85 | 23.78 | 29.81 | 36.87 | 7.59 | 13.52 | 19.56 | 26.62 |
| | DinoV2(s) | 10.64 | 16.39 | 22.54 | 28.71 | 38.02 | 5.74 | 11.89 | 18.06 | 27.37 |
| | Resnet18-Quant | 5.83 | 9.07 | 12.20 | 15.93 | 21.00 | 3.24 | 6.38 | 10.10 | 15.17 |
| TinyImagenet | Resnet18-ReLU | 5.51 | 9.58 | 14.90 | 19.13 | 21.49 | 4.07 | 9.39 | 13.62 | 15.97 |
| | DinoV2(s) | 9.39 | 14.73 | 23.30 | 33.78 | 42.52 | 5.34 | 13.91 | 24.39 | 33.13 |
| | Resnet18-Quant | 3.27 | 4.86 | 7.26 | 10.43 | 13.18 | 1.59 | 3.99 | 7.16 | 9.90 |

Table 2: Measuring the drop in accuracy

| | | 2→3 | 2→4 | 2→5 | 3→4 | 3→5 | 4→5 |
|---|---|---|---|---|---|---|---|
| Dataset | Model/Method | | | | | | |
| CIFAR10 | Resnet18-ReLU | 5.70 | 13.25 | 23.92 | 7.55 | 18.22 | 10.67 |
| | DinoV2(s) | 5.08 | 10.93 | 20.28 | 5.85 | 15.20 | 9.35 |
| | Resnet18-Quant | 2.34 | 5.54 | 9.98 | 3.20 | 7.64 | 4.45 |
| CIFAR100 | Resnet18-ReLU | 5.93 | 11.97 | 19.03 | 6.04 | 13.10 | 7.06 |
| | DinoV2(s) | 6.15 | 12.32 | 21.63 | 6.17 | 15.48 | 9.31 |
| | Resnet18-Quant | 3.13 | 6.86 | 11.92 | 3.73 | 8.79 | 5.06 |
| TinyImagenet | Resnet18-ReLU | 5.32 | 9.55 | 11.90 | 4.23 | 6.58 | 2.35 |
| | DinoV2(s) | 8.57 | 19.05 | 27.79 | 10.48 | 19.22 | 8.74 |
| | Resnet18-Quant | 2.40 | 5.56 | 8.31 | 3.17 | 5.92 | 2.75 |

Table 3: Measuring the drop in accuracy (contd. from table 2)

**Measuring Robustness of QAct:** To measure the robustness of the proposed method we use the metric `Acc@Dist_i - Acc@Dist_j` which measures the drop in accuracy when the distortion severity is increased from $i \to j$. A method is considered to be better if the values are *lower*, i.e if the drop in accuracy is smaller than comparitive methods. Tables 2 and 3 shows the comparison between ReLU, QAct and DinoV2(s). We see that, in all the cases QAct outperforms both ReLU and DinoV2(s) at all possible $i \to j$.

## D  Watershed Loss

The authors in (Challa et al., 2024a) proposed a novel classifier – *watershed classifier*, which works by learning similarities instead of the boundaries. Below we give the brief idea of the loss function, and refer the reader to the original paper for further details.

1. Let $(\boldsymbol{x}_i, y_i)$ denote the samples in each batch, and let $f_\theta$ denote the embedding network. $f_\theta(\boldsymbol{x}_i)$ denotes the corresponding embedding.
2. Starting from randomly selected seeds in the batch, propagate the labels to all the samples. Let $\hat{y}_i$ denote the estimated samples. For each $f_\theta(\boldsymbol{x}_i)$ and for each label $l$, obtain the nearest neighbour in the samples in the set,

$$\mathcal{S}_l = \{f_\theta(\boldsymbol{x}_i) \mid \hat{y}_i = y_i = l\} \tag{11}$$

that is, all the samples of class $l$ labelled correctly. Denote this nearest neighbour using $f_\theta(\boldsymbol{x}_{i,l,1nn})$.
3. Then the loss is given by,

$$\text{Watershed Loss} = \frac{-1}{n_{\text{samples}}} \sum_{i=1}^{n_{\text{samples}}} \sum_{l=1}^{L} I[y_i = l] \log \left( \frac{\exp\left(-\|f_\theta(\boldsymbol{x}_i) - f_\theta(\boldsymbol{x}_{i,l,1nn})\|\right)}{\sum_{j=1}^{L} \exp\left(-\|f_\theta(\boldsymbol{x}_i) - f_\theta(\boldsymbol{x}_{i,j,1nn})\|\right)} \right) \tag{12}$$

**Why Watershed Loss?:** Observe that the loss in equation 12 implicitly learns representations consistent with the RBF kernel, which is known to be translation invariant. Minimizing this loss function, hence, will learn translation invariant kernels. This is important for obtaining networks robust to distortions.

If one uses (say) cross-entropy loss, then the features learned would be such that the classes are linearly separable. Contrast this with watershed, which instead learns a similarity between two points in a translation invariant manner.

**Remark:** Observe that the watershed loss is very similar to metric learning losses. The authors in (Challa et al., 2024a) claim that this offers better generalization, and show that this is consistent with 1NN classifier. Moreover, they show that this classifier (without considering $f_\theta$) has a VC dimension which is equal to the number of classes. While metric learning losses are similar, there is no such guarantee with respect to classification. This motivated our choice of using watershed loss over other metric learning losses.

## E    Formal Explanation of Failure Mode in Linear Models and why Quantiles fix it

Here, we discuss the failure mode presented in section 1 formally in the context of linear models. And then show, why the quantile activation as presented corrects this.

**Linear Models under simple Distribution Shift:** Let $\{(\boldsymbol{x}_i, y_i)\}$ denote the set of examples from the joint probability distribution of $p(\mathrm{x}, \mathrm{y})$. Assume for simplicity the binary classification problem i.e $y_i \in \{0, 1\}$. Let $w^t \boldsymbol{x} + b$ denote the linear classifier trained on this. Note that the classification rule is then simply $I[w^t \boldsymbol{x} + b \geq 0]$.

Now let $\boldsymbol{A}$ denote a transformation matrix. Let $\mathbf{x}$ (marginal) be transformed as $\boldsymbol{A}\mathbf{x}$. If $\boldsymbol{x}' = \boldsymbol{A}\boldsymbol{x}$ denote a sample from the transformed distribution, the classification rule applied to this sample is

$$I[w^t \boldsymbol{x}' + b \geq 0] = I[w^t(\boldsymbol{A}\boldsymbol{x}) + b \geq 0] \tag{13}$$

It is easy to see that, even under the simple case of $\boldsymbol{A} = c\boldsymbol{I}$, the class assignment can vary arbitrarily with $c$.

**Linear Models + Quantile Activation under simple Distribution Shift:** What we propose in the article, is to instead consider the following classification rule. If $x' = \boldsymbol{A}x$ denote a sample from the transformed distribution,

$$P(w^t \boldsymbol{x}' > w^t(\boldsymbol{A}\mathbf{x})) \geq 0.5 = P(w^t(\boldsymbol{A}\boldsymbol{x}) > w^t(\boldsymbol{A}\mathbf{x})) \geq 0.5 \tag{14}$$

Here, if $\boldsymbol{A} = c\boldsymbol{I}$, then it is easy to see that the class assignment is *independent of the value of c*. More formally, we have the following proposition.

**Proposition 1.** *Let A denote the transformation as described above. Then, as long as $\boldsymbol{A}^t w = \alpha w$ with $\alpha > 0$ then the class assignment of the linear model with quantile activation does not change under the transformation $\boldsymbol{A}\boldsymbol{x} + z$ where z can be arbitrary.*

The proof follows by simple substitution in equation 14.

