# OpenReview forum: "Quantile Activation: Correcting a failure mode of traditional ML models"
_TMLR — Accepted by TMLR_

### Review · Reviewer_oSZT · 2025-02-06

**Summary Of Contributions:**

The paper studies modifications of neural net architectures that improve robustness against domain shift. The paper proposes a novel "Quantile Activation" function for single neurons that operates on batch level. By operating on batch-level (and assuming that batch-elements come from the same distribution), the proposed quantile activation function can increase robustness to certain type of domain shifts. In experimental results, the method is shown to perform well on common corruption benchmarks (CIFAR10C, CIFAR100C, TinyImagenetC) in terms of accuracy and calibration.

**Audience:**

Yes

**Broader Impact Concerns:**

no concerns

**Claims And Evidence:**

No

**Requested Changes:**

The paper needs to be improved in terms of clarity of problem statements, motivation of the studied problem, identification of most closely related work, and experimental evaluation (comparison to simple batch-level baselines, study of domain shifts beyond just common corruptions). I believe a major revision is required.

**Strengths And Weaknesses:**

Strengths:
 * the idea of using quantile-based neural networks is promising as it does not require strong assumptions about the distribution
 * robustness to domain shifts is a very relevant research field that is of interest to TMLR's audience

Weaknesses:
  * The type of domain shift is not clearly defined. The motivating examples are focused on concept shift/drift, where p(Y|X) changes. However, the experiments on common corruptions are actually examples of covariate shift, that is: the feature distribution p(X) changes over time. The paper would benefit from more rigor in defining the problem statement and the assumptions on changes in data distribution
 * An underlying assumption of the work seems to be that all elements in a batch come from the same distribution (and distribution shift only happens between different batches). This is a strong assumption, which should be motivated better
 * Under the assumptions that all elements of a batch come from the same distribution, there are many alternative approaches and baselines that could be compared to, for instance having a normalization layer operate across batch elements (like batch normalization does at training time)
 * Related work should be more focused on "Batch-wise Domain Adaptation"
 * Experimental evaluations are too narrow, focusing exclusively on "common corruption" domain shift.
 * Figure 1 right is confusing and would require more explanation

---

> ### Author Response · Authors · 2025-02-08
>
> We thank the reviewer for the comments and are grateful for the insights provided.
>
> However, we believe there is a miscommunication regarding the aim and scope of the article. Specifically, the aim of the article is more general than addressing distribution shift. Any feedback regarding how to highlight this is highly appreciated.
> 1. The main **aim** of the article is to fix the failure mode of traditional ML model.
> 2. *What is the failure mode?-* Recall that the key assumptions of traditional ML models is that there is a underlying fixed distribution $p(x,y)$ and one hopes to find a *best fit* within the hypothesis class $\mathcal{H}= \lbrace f : \mathcal{X} \to \mathcal{Y}\rbrace$   However, data in general is usually not from a fixed distribution, and hence any class of *fixed* functions cannot learn in this case.
> 3. *Examples of failure mode?*
> 	- The toy example illustrates a case where each batch of samples comes from a different distribution. In this extreme example, the Bayes error of the combined distribution is 0.5 and hence **no** traditional ML hypothesis classes can learn this.
> 	- Another example of the failure mode is that of classic distribution (covariate) shift.
> 4.  *How to fix the above failure mode?* The main idea is to design a class of functions which are neural networks and which *adapt to the distribution* for learning.  One can think of the quantile activations as a functional $\phi(F,x) = F(x)$ where $F$ is a cumulative distribution function (cdf). We refer to $F$ as a *context distribution*. (Note that the above definition of quantile activation is not exactly true since we also force the median to be $0$.)
> 5. *Scope of the article:* Our aim is to present the above novel quantile activation and evidence it's usage. The aim is not to obtain state-of-the-art results on distribution shift.
> 	- Note that the above framework is a more general than dealing with distribution shifts alone.
> 	- Batchnorm (and adaptive versions of it) can also be thought of as functionals $\psi(X,x) = (x - E[X])/Stdev(X)$ but the entire distribution is restricted to just it's expectation and variance. Thus quantile activation is a lot more general than Batchnorm.
> 	- One can easily construct several examples where adaptive batchnorm (and it's variants) fail. Let $p_{0}, p_1$ denote the class distributions of classes $0$ and $1$ respectfully. Say the training is on the mixture $0.9p_0 + 0.1p_1$ while testing is on the mixture $0.5p_0 + 0.5p_1$. Here, batchnorm would fail since the boundary will be shifted. Quantile activation on the other hand naturally reweighs the classes to preserve the boundary. It is widely known that median is a better *robust statistic* than mean.
>
> **concept shift vs covariate shift:** The examples indicate covariate shift and not concept shift. Note $p(x,y) = p(y/x)p(x)$. Note that if we have a covariate shift $p(x) \to p'(x)$, then $p(x,y)$ also changes even if $p(y/x)$ remains the same.
>
> **Assumption that all samples in a batch are from same distribution is strong** We respectfully disagree. Standard ML models assume the entire dataset to be from the same distribution. Here we only assume each batch of sample is from the same distribution which is a lot weaker.
>
> **Batch Norm is sufficient:** As we discuss above, quantile activation is a lot more general than simple batch norm. For example, quantile activation can handle imbalanced distribution shifts too. Moreover, median/quantiles are known to be more robust than averages.
>
> **Figure 1 is confusing:** As the caption states - *Histogram of accuracy over 1000 different combinations of $\mu_1$, $\mu_2$  for both ReLU activation and after incorporating QAct* -- That is, (i) we sample  $\mu_1, \mu_2$ (ii) obtain the accuracy for this (iii) Repeat (i) and (ii) several times to get a list of accuracies (iv) Plot the kernel density estimate of these accuracies for both ReLU and QAct. Please let us know if we can make this better.
>
> **Batch-Level baselines:** We hope to include the results from [7], [3], [4] and [6]. We  could not find the code for [2],[5] to reproduce those results. The initial results (directly taken from the article) for cifar10c are given below. Observe that Quantile activation is highly competitive.
>
> | Method                          | Drop in accuracy $1 \to 5$ |     |
> | ------------------------------- | -------------------------- | --- |
> | Resnet18+Quant                  | 12.86                      |     |
> | Resnet26+Adaptive Batch Norm[7] | 20.8                       |     |
> | Resnet26 + MEMO [3]             | 17.55                      |     |
> | Resnet18+Augmix[4]              | 14.4                       |     |
> | WRN40-4+AFN[6]                  | 12.3                       |     |
> |                                 |                            |     |

---

> > ### Author Response · Authors · 2025-02-08
> >
> > **(requested changes) The paper needs to be improved in terms of clarity of problem statements, motivation of the studied problem, identification of most closely related work, and experimental evaluation (comparison to simple batch-level baselines, study of domain shifts beyond just common corruptions)**
> >
> > - We believe that the problem statement and it's motivation is indeed clear. Please let us know the main issues so that we can correct it in the revision.
> > - We plan to include the references below and also include the results for which the code is available.
> > - The aim of the experiments is to provide evidence that quantile activation corrects covariate shift, and hence we only considered common corruptions. Note that common corruptions is actually a set of **15 different distribution shifts** and hence believe that this is sufficient evidence to corroborate out claims.
> >
> > [1] Li et al., "Revisiting Batch Normalization For Practical Domain Adaptation", 2016
> > [2] Schneider et al., "Improving robustness against common corruptions by covariate shift adaptation", 2020
> > [3] Zhang et al., "MEMO: Test Time Robustness via Adaptation and Augmentation", 2022
> > [4] Hendrycks et al., "AUGMIX: A SIMPLE DATA PROCESSING METHOD TO IMPROVE ROBUSTNESS AND UNCERTAINTY", 2020
> > [5]Schwinn et al., "Improving Robustness against Real-World and Worst-Case Distribution Shifts through Decision Region Quantification", 2022
> > [6] Zhou et al., "AFN: ADAPTIVE FUSION NORMALISATION VIA AN ENCODER-DECODER FRAMEWORK", 2024
> > [7] Mirza et. al., "The Norm Must Go On: Dynamic Unsupervised Domain Adaptation by Normalization" 2022

---

> > > ### Comment · Reviewer_oSZT · 2025-02-10
> > > **Feedback**
> > >
> > > Thanks for including a subset of relevant work from the field to test-time adaptation and source-free domain adaptation in your response - that was missing from the original submission and why I requested an "identification of most closely related work". That is a good starting point for an actual "related works" section that discusses how this work relates to prior work, how assumptions differ, what advantages and disadvantages different approaches have, and why assumptions being made in this work are of interest to TMLR's audience.
> > > The second part is to provide empirical evidence that the proposed work actually provides the presumed advantages compared to related work (the "traditional ML model" which seems to correspond to iid assumptions on train/test data is not a sufficient baseline for empirical evaluations).
> > > Just to repeat: I think the work on quantile activation is promising and the problem addressed is relevant - but the paper lacks grounding in the related works and assumptions such as batches coming from the same domain are too restrictive.

---

> > > > ### Author Response · Authors · 2025-02-11
> > > >
> > > > We thank the reviewer for the response. As per the reviewer's suggestions we shall include the above discussion in the updated pdf (as per recommendation this is submitted after all the three reviews) with the new baselines and relations with existing literature on distribution shift.
> > > >
> > > > To address the two main issues flagged by the reviewer, we shall also include the following lines
> > > >
> > > > **... but the paper lacks grounding in the related works**
> > > >
> > > > **Relation to existing literature on Distribution Shift:** *Adaptive Batch Normalization and its variants ([1], [2], [6], [7]) can be interpreted as special cases of context distribution modification, where only the mean and variance are adjusted. In contrast, methods that modify model parameters ([3]) or alter the input distribution ([5]) preserve the pre-trained classifier but introduce a computationally expensive optimization step during inference. Empirical results indicate that adapting batch normalization statistics is at least as effective, if not superior, to these approaches. Furthermore, augmentation strategies ([4]) are advantageous when domain-specific invariances are known, but such knowledge is not always available*
> > > >
> > > > **... and assumptions such as batches coming from the same domain are too restrictive.**
> > > >
> > > > *Note that the batch-size can actually be as small as $1$ (and augmentations could increase it as done in [3]). Moreover, one can also use simple bootstrap equivalent of updating the statistics with a single sample (as done in [2]). However, as observed in the references, accuracies are better if there is more diversity in the batches and more unlabelled samples are used.*

---

### Review · Reviewer_NoTz · 2025-02-07

**Summary Of Contributions:**

The authors demonstrate that current ML frameworks (specifically DL) are not able to handle distribution shifts of the data at train and test times. They propose the quantile activation function (QAct), which uses batch information to handle distribution shifts online. Namely, the activation function computes the activation for the entire batch of samples and adjusts the activation of individual samples based on this value.
In an empirical study they demonstrate that their method improves robustness to distribution shifts on several datasets.

**Audience:**

Yes

**Claims And Evidence:**

Yes

**Requested Changes:**

- Improve the presentation of the paper and transitions between talking points. At the moment the introduction is mostly a list of discussion points that follow after another (I acknowledge that there already is some structure but IMO it could be improved to make the paper easier to understand).
- Considerably extend the discussion of related work and back up all claims made in the paper by references or experiments
- Add baselines from other works that address the same problem (sometimes even using batch statistics)
- Add discussion how this approach may be applied in practice (since batch statistics are not always available)

**Strengths And Weaknesses:**

**Strengths**

- The paper addresses a relevant problem of the TMLR community
- The proposed approach is practical, e.g., it can be applied to all scenarios where data is obtained in batches or sequentially
- The properties of the method are evaluated in several experiments (however, they are not compared to similar approaches)

**Weaknessess**

- Overall presentation of the paper and writing requires improvement (e.g., lack of motivation, large number of paragraphs without clear transitions between them)
- No baselines that try to tackle the problem of OOD generalizations except for DINOv2. However, there are more suitable baselines (see other references)
- Discussion of related work and overall references are not extensive enough
- Add references to the first paragraphs of the introduction
- There are several works that use batch information at test time (batch statistics) to improve robustness (e.g., [1,2]). The authors should frame their contribution accordingly and also compare against these works. Some works improve robustness without relying on batch statistics [3, 4] (not necessary to compare to those but still interesting)
- Would have been interesting to simulate the proposed approach in a real-world use case where data is obtained over time (e.g., activation statistics are running averages)


[1] Li et al., "Revisiting Batch Normalization For Practical Domain Adaptation", 2016
[2] Schneider et al., "Improving robustness against common corruptions by covariate shift adaptation", 2020
[3] Zhang et al., "MEMO: Test Time Robustness via Adaptation and Augmentation", 2022
[4] Schwinn et al., "Improving Robustness against Real-World and Worst-Case Distribution Shifts through Decision Region Quantification", 2022

---

> ### Author Response · Authors · 2025-02-08
>
> We thank the reviewer for the comments and are grateful for the insights and references provided.
>
> We also appreciate the comment on the presentation, since we believe there is a miscommunication regarding the aim and scope of the article. Specifically, the aim of the article is more general than addressing distribution shift. Any feedback regarding how to highlight would be helpful.
>
> 1. The main **aim** of the article is to fix the failure mode of traditional ML model.
> 2. *What is the failure mode?-* Recall that the key assumptions of traditional ML models is that there is a underlying fixed distribution $p(x,y)$ and one hopes to find a *best fit* within the hypothesis class $\mathcal{H}= \lbrace f : \mathcal{X} \to \mathcal{Y}\rbrace$   However, data in general is usually not from a fixed distribution, and hence any class of *fixed* functions cannot learn in this case.
> 3. *Examples of failure mode?*
> 	- The toy example illustrates a case where each batch of samples comes from a different distribution. In this extreme example, the Bayes error of the combined distribution is 0.5 and hence **no** traditional ML hypothesis classes can learn this.
> 	- Another example of the failure mode is that of classic distribution (covariate) shift.
> 4.  *How to fix the above failure mode?* The main idea is to design a class of functions which are neural networks and which *adapt to the distribution* for learning.  One can think of the quantile activations as a functional $\phi(F,x) = F(x)$ where $F$ is a cumulative distribution function (cdf). We refer to $F$ as a *context distribution*. (Note that the above definition of quantile activation is not exactly true since we also force the median to be $0$.)
> 5. *Scope of the article:* Our aim is to present the above novel quantile activation and evidence it's usage. The aim is not to obtain state-of-the-art results on distribution shift.
> 	- Note that the above framework is a more general than dealing with distribution shifts alone.
> 	- Batchnorm (and adaptive versions of it) can also be thought of as functionals $\psi(X,x) = (x - E[X])/Stdev(X)$ but the entire distribution is restricted to just it's expectation and variance. Thus quantile activation is a lot more general than Batchnorm.
> 	- One can easily construct several examples where adaptive batchnorm (and it's variants) fail. Let $p_{0}, p_1$ denote the class distributions of classes $0$ and $1$ respectfully. Say the training is on the mixture $0.9p_0 + 0.1p_1$ while testing is on the mixture $0.5p_0 + 0.5p_1$. Here, batchnorm would fail since the boundary will be shifted. Quantile activation on the other hand naturally reweighs the classes to preserve the boundary. It is widely known that median is a better *robust statistic* than mean.
>
>
> **Baselines (New):**  We hope to include the results from [7], [3], [4] and [6]. We  could not find the code for [2],[5] to reproduce those results. The initial results (directly taken from the article) for cifar10c are given below. Observe that Quantile activation is highly competitive.
>
> | Method                          | Drop in accuracy $1 \to 5$ |     |     |
> | ------------------------------- | -------------------------- | --- | --- |
> | Resnet18+Quant                  | 12.86                      |     |     |
> | Resnet26+Adaptive Batch Norm[7] | 20.8                       |     |     |
> | Resnet26 + MEMO [3]             | 17.55                      |     |     |
> | Resnet18+Augmix[4]              | 14.4                       |     |     |
> | WRN40-4+AFN[6]                  | 12.3                       |     |     |
> |                                 |                            |     |     |
> **Presentation and related changes:**  (as per recommendation this is done after the three reviews are received)
> - We shall upload the revised draft which includes the references/results as suggested.
> - Also will include a discussion comparing our approach with others.
> - Include cross-references to the experiments to support the statement within the text.
> - We shall include the detailed tables of above baselines.
>
> [1] Li et al., "Revisiting Batch Normalization For Practical Domain Adaptation", 2016
> [2] Schneider et al., "Improving robustness against common corruptions by covariate shift adaptation", 2020
> [3] Zhang et al., "MEMO: Test Time Robustness via Adaptation and Augmentation", 2022
> [4] Hendrycks et al., "AUGMIX: A SIMPLE DATA PROCESSING METHOD TO IMPROVE ROBUSTNESS AND UNCERTAINTY", 2020
> [5]Schwinn et al., "Improving Robustness against Real-World and Worst-Case Distribution Shifts through Decision Region Quantification", 2022
> [6] Zhou et al., "AFN: ADAPTIVE FUSION NORMALISATION VIA AN ENCODER-DECODER FRAMEWORK", 2024
> [7] Mirza et. al., "The Norm Must Go On: Dynamic Unsupervised Domain Adaptation by Normalization" 2022

---

> > ### Comment · Reviewer_NoTz · 2025-02-08
> > **Thanks for the detailed response**
> >
> > I thank the authors for the clarifications and additional experiments.
> > The stated changes address my concerns. I will review the updated version once the other reviews have been answered

---

> > > ### Comment · Reviewer_NoTz · 2025-02-26
> > > **Updated manuscript**
> > >
> > > I thank the authors for the changes. The related work section was considerably improved and the new experiments underline the effectivness of the proposed approach. The concerns of the other reviewers were mostly addressed as well. I do not request any further changes.

---

### Review · Reviewer_MssE · 2025-02-18

**Summary Of Contributions:**

In this article, the authors propose to use QAct, a new activation function for neural network layers, for performing predictions that are more robust to dataset distortions. As QAct is computed as a quantile (corresponding to the current pre-activation value) of the distribution of all pre-activation values in the batch, its gradient (and thus its incorporation into autodifferentiable models) is equal to the density of the distribution, which can be obtained with kernel density estimators. The authors then provide numerical experiments illustrating the robustness of neural net architectures that are augmented with QAct against several degrees of severity of distortions and across several datasets. In particular, positive comparison with respect to Dino-V2, a state-of-the-art model in domain generalization, is emphasized.

**Audience:**

Yes

**Claims And Evidence:**

No

**Requested Changes:**

---The connection with the work of (Challa et al., 2023) is unclear. Is the proposed approach a simple extension, or does it propose a radically new idea within this framework? The link between the first two paragraphs of Section 2 is not evident at all. I probably missed something, but the key point of the method of (Challa et al., 2023) is to refit the model at different quantiles, in order to appropriately capture the data shape, and thus to be robust to data distortions. In the proposed approach, I don't see the same happening: the model stays the same across batches. Hence, I find the transition between QuantProb and QAct difficult to make.

---The experiments presented in the introduction looks nice, but they are only briefly discussed, and there are no additional details about them after the method has been presented. In particular, I would have really liked to see why QAct allows to obtain the results in Figure 1 & 2, as it is not clear just from its definition. Indeed, taking, e.g., Figure 1, I do not see why QAct allows to get better accuracy: it is possible that, from a batch to another, the projections of the Gaussian of mu_1 and that of the Gaussian of mu_2 switch positions, and thus using QAct will not help in separating them (in one batch, points from mu_1, resp. mu_2, will be in the left, resp. right, part of the distribution, and on the right, resp. left, part in another batch). I am sure that I misunderstood something, so explaining these examples again in much greater details in section 4 would be appreciated. Also, what are the colored lines in Figure 1a? I think they are not explained in the text.

---I do not understand why Algorithm 1 requires to approximate the quantile of QAct with a list of predefined quantiles tau_1,...,tau_n: why not computing the quantile exactly (using, e.g., a function like numpy.quantile)? This could introduce significant biases if n_tau is too small and/or not uniform across [0,1] (such as, e.g., [0.70, 0.75, 0.80]).

---The paragraph "Grounding the Neurons" is unclear and contradictory with the main argument of the paper (at first sight). It looks like the fact that QAct always outputs a uniform distribution between [0,1] is problematic, but (1) Figure 3 seems to indicate that it is actually a desired property, and (2) I don't see why the proposed transformation (that forces the median to be 0) will help. This could be better explained.

---There are many factors that could be discussed in experiments: what is the dependence of the performance with respect to the batch sizes? To the kernel bandwidth in the KDE (assuming it is a Gaussian kernel)? To the number of samples needed to compute the KDE?

**Strengths And Weaknesses:**

The experiments, even though they are proof-of-concepts, look pretty promising and interesting. I think the paper is also based on a natural, yet potentially powerful idea. However, I am quite concerned about the writing: in its current shape, I had a lot of difficulties following the flow of the article, and many questions remain, that, I think, are important to address (see below).

---

> ### Author Response · Authors · 2025-02-19
>
> We thank the reviewer for the comments and are grateful for the insightful questions.
>
> **Regarding Presentation:** We tried to improve the writing of the article by adding in a few connections between paragraphs. Please do let us know if any further improvements can be made with respect to exposition.
>
> **Connection with (Challa et al., 2023)**  We agree with the statement *"key point of the method of (Challa et al., 2023) is to refit the model at different quantiles, in order to appropriately capture the data shape"*. Further to this, our main takeaway (in simple words) from (Challa et al., 2023) is -- Computing the relative quantiles is akin to obtaining probabilities (as their experiments/theory showed). Moreover, the authors also managed to get this information from simple thresholding (step 2 in section 2 in our article).
>
> However the authors of that article missed 2 important scope for improvements: (i) You do not need to give the quantile as an input, but can just use the quantile of the pre-activations instead. (ii)  One should perform their operation at every neuron level and not just at input/output as (Challa et al., 2023) did. However, correcting these two are non-trivial which is the contribution of this article.
>
> These allowed us to obtain better generalisation w.r.t accuracies. Note that the  (Challa et al., 2023) only claims improvement w.r.t calibration not accuracies.
>
> **Figure 1 explanation:** There is indeed a subtlelity in figure 1, which we did not go into since this might cause confusion. Mainly both the following statements are true empirically:
> 1. If instead of taking $\mu_1, \mu_2$ on a disk we take them to be either $\mu_1, \mu_2 = ((0,1),(-1/2,\sqrt{3}/2)$ or $\mu_1, \mu_2 = ((-1/2,\sqrt{3}/2),(0,1))$ quantile activation will not be able to learn. This corresponds to flipping the labels or semantic shift.
> 2. On the other hand, as in figure1 we take $\mu_1,\mu_2$ in a disk then quantile activation can learn. But the key thing to note is the switch in the relative positions of $\mu_1,\mu_2$ happens after rotating $180^{\circ}$
>
> The main difference between them is - (1) corresponds to semantic shift where the labels change while (2) corresponds to covariate shift.
>
> The intuition is -- In figure 1 because of the change in directions of the optimal boundary (dotted lines) there is definitely a singularity of the learned function, somewhere in the unit disk. Now, relu activation being a continuous function by definition cannot learn this. Quantile activation can resolve the covariate shifts as illustrated in figure 1, but cannot handle label (semantic) shifts.
>
> **Why Algorithm 1 requires to approximate the quantile of QAct with a list of predefined quantiles:** Sorry for the confusion. But we do perform weighted quantile computation exactly. Because we had to perform the weighted version, we could not use torch.quantile directly and instead implemented our own on the GPU. Please see the function weighted_quantile_vectorized [here](https://anonymous.4open.science/r/QuantAct-534C/src/quantile_activation_with_kde.py). We sample only for KDE for computational efficiency.
>
> Regarding the uniformity of quantiles -- we agree that if the quantiles are not uniformly spaced in $[0,1]$ it poses a problem. However, since this is an hyperparameter, we can fix them to be uniformly spaced in $[0,1]$ without effecting the rest of the pipeline.
>
> **Why Grounding the Neurons?** We agree that the main improvement (w.r.t distribution shift) is because convert the arbitrary distributions into uniform distributions. However, the naive non-weighted conversion of distribution to uniform cannot learn the discriminatory features with gradient descent. The issue here is more about training with gradient descent. It is possible that some intermediate discriminatory features are not equally present in the distribution.
>
> To understand this better, consider the following - Let $f$ denote a discriminatory feature which is to be learned but is present in only 25% of the population. However, if the linear model (a.k.a neuron) has to learn $f$ is there (class 0) or not (class 1), it should automatically account for that imbalance. Gradient descent without weighting is known to fail here. We overcome this problem by *giving the weights such that the median is equal to 0*.  ( We include this in the revised draft.)
>
> **Batch size and kernel width:** We include a discussion on the importance of batch size and methods to side-step this issue in the revised draft. The overarching principle being -- The context distribution should match the ground-truth. For kernel density estimate we simply use the rule from [1]. Since we also use a batch-norm before quantile activation, using the rule we get the bandwidth to be $\approx 0.27$
>
> [1] Silverman, B. W. (2018). _Density estimation for statistics and data analysis_. Routledge.

---

> > ### Comment · Reviewer_MssE · 2025-03-07
> >
> > Thank you for your comments.
> >
> > **About Figure 1:** I still don't get it. Is QAct working better because the neuron pre-activations correspond more or less to the angle of the data points along the circle, and thus adding 180 degrees to both $\mu_1$ and $\mu_2$ does not change the quantiles? But in that case, couldn't you get in trouble if you use angles modulo $2\pi$?
> >
> > **About quantile computation in Algorithm 1:** my question was rather about the fact that I have the impression that you could avoid using a list of pre-defined quantiles entirely. What prevents you from computing `torch.quantile` on the weighted distribution directly?
> >
> > **About grounding:** what do you mean by $f$ is there or not? Do you consider $f$ to be "present" (resp, "missing") if it is positive (resp. negative)? And do you force the median to be zero so that the input distribution looks more uniform?
> >
> > **About batch size:** my comment was more about what is the dependence of performances w.r.t. the batch size? Empirically, is QAct leading to better accuracies when batches are bigger (because the quantiles are more precise)?

---

> ### Author Response · Authors · 2025-03-08
>
> **About Figure 1:** *Is QAct working better because the neuron pre-activations correspond more or less to the angle of the data points along the circle, and thus adding 180 degrees to both $\mu_1, \mu_2$ does not change the quantiles?*
>
> Yes and No. Let us explain this in a different way. Do let us know if the following explanation clarifies the working of QAct.
>
> For the time being, let's move away from figure 1. Let $\{x_i, y_i\}$ denote the samples where $x_i \in \mathbb{R}^d$ ($d >> 2$)  and $y_i \in \{ 0,1 \} $. Let $w^t x + b$ denote the classifier which separates the classes perfectly. The question we ask is -- Under what transformations $x \to Ax$ does the classifier preserve it's accuracy?
>
> As we discuss in appendix E, for standard linear models even if $A = c \times Id$ would not preserve the accuracy. On the other hand, for quantiles if $A^t w = \alpha w$ ($\alpha > 0$), then it preserves the accuracy.  So, the class of matrices for which the accuracy is preserved is nothing but all *completions* of the matrix $A$ such that $A^tw = \alpha w$. (Remark: Please note that in eq(14), Appendix E we use $x$ for the vectors and $\textnormal{x}$ for the distribution)
>
> Coming back to figure 1 - Think of a neural network as projecting the 2d points into high-dimensions (in a possibly non-linear way). If the projection to high dimensions is such that the rotation in 2d preserves $A^t w = \alpha w$ in the high dimension for the final classification, then the accuracy would be preserved. Quantile activations implicitly do this.
>
> The important thing to note is -- The projection to the high dimensions, and the classifier in the subsequent layer, are *all* learned using SGD and none of these behaviours are hardcoded.
>
> **About quantile computation in Algorithm 1** Sorry again, but we do not think we understand the question. Can you please tell us a little more?
>
> Is the question -- In the forward pass we are computing $\sum_i I[x \geq q_i]$ where $q_i$ is the weighted quantile, but we can directly do $\sum_i I[x \geq x_i]$ where $x_i$ is the sample from weighted distribution?
>
> On a CPU usually this would not matter. However, on a GPU we aim to exploit the parallelism as much as possible.
> - In $\sum_i I[x \geq x_i]$ the total number of summands (i.e $I[x \geq x_i]$) we need to remember is $nSamples \times nSamples$ which consumes a lot of memory. Especially for a image input after a convolution layer we would have approximately $B \times H \times W$ samples. This implies the memory requirement of approximately $100k \times 100k$
> - In  $\sum_i I[x \geq q_i]$ this is $nSamples \times nQuantiles$. And fixing the quantiles to be $100$ or $1000$ allows us to balance the memory requirement and accuracy better.
>
> **About grounding:**
> - what do you mean by f is there or not? Do you consider f to be "present" (resp, "missing") if it is positive (resp. negative)?
>
> Yes. We assume that $f$ is present if it is positive and missing if it is negative.
>
> -  And do you force the median to be zero so that the input distribution looks more uniform?
>
> We would rather state it as - "We force the median to be zero so that the input distribution (to the neuron) looks more *balanced* for the backward pass". The balance requirement is for gradient based learning.
>
> **About batch size:** Yes. Lower batch sizes (assuming you are not using any other tricks) does give lower accuracies.
>
> We have not done an empirical study on this explicitly since it is already widely established that the quantile estimates become better with more number of samples [1].
>
> [1] Koenker, R. (2005). Quantile regression (Vol. 38). Cambridge university press.

---

> > ### Comment · Reviewer_MssE · 2025-03-08
> >
> > Thank you for these detailed answers!! I have no remaining questions. My suggestion would just be to add those answers to the main text, they really helped me figuring out what was happening.

---

> > > ### Author Response · Authors · 2025-03-08
> > >
> > > We thank you for the questions and comments. We agree that the explanations would add more value to the article. We shall add these to the main text consolidating all the other comments as well.

---

### Author Response · Authors · 2025-02-19

We thank the reviewers for their valuable comments and incorporated the changes in to the latest version of the pdf.

Please note that we have uploaded an updated pdf where the suggested changes have been highlighted in blue.

---

### Decision · Action_Editor_4PXk · 2025-03-21

**Recommendation:** Accept as is

**Comment:**

The reviewers generally agree that the paper presents an interesting idea and addresses a relevant problem for the TMLR community. One point raised by the reviewers is that the paper could be improved by expanding the experimental evaluation to include a wider range of domain shifts and by providing more in-depth intuitions behind the method's effectiveness. However, the current version of the manuscript is sound and presents a novel approach with sufficient empirical evidence to warrant publication in TMLR.  The core idea of QAct is interesting and the experimental results, while limited, are convincing within their scope.  Further enriching the paper with more extensive experiments and deeper intuitions would undoubtedly make it stronger, but requesting a major revision at this stage, given the positive assessments from the majority of reviewers and the authors' revisions, would likely be inefficient and potentially a waste of both reviewer and author time.

**Audience:**

The paper addresses a relevant problem in machine learning – the failure of standard models under distribution shifts – and proposes a novel and practical solution in the form of the Quantile Activation function. The TMLR audience, which includes researchers and practitioners interested in robust and reliable machine learning, would find the proposed method and its empirical evaluation relevant and potentially valuable.

**Claims And Evidence:**

While some reviewers initially had concerns about the breadth and depth of the experimental evaluation, the authors have addressed most of these concerns in their revisions and responses. The application of QAct to improve robustness to common corruptions has been demonstrated it in common benchmarks, even if further experiments could strengthen the evidence. The provided ablations and analysis could be useful for people trying to reproduce these findings.

---

> ### Author Response · Authors · 2025-03-25
> **Thanks**
>
> We thank the reviewers for the detailed reviews and the time to engage with us. We feel optimistic about the paper. Thanks a lot AE!